# Beyond Correctness: Distance-Based Social Dynamics of Multi-Agent Debate

Seungwoong Ha [1]    Melanie Mitchell [1]

## Abstract

Multi-agent debate (MAD) systems are often evaluated using binary correctness or peer agreement, yet such evaluations obscure how individual agents revise their answers during social interaction. We study the microscopic dynamics of answer revision in large language models (LLMs) using ConceptARC, a 2D grid-reasoning benchmark that admits quantitative distance measures between candidate solutions. By exposing a target model to controlled configurations of peer answers, we analyze how the likelihood and direction of revision depend on both social context and the distance between answers and the ground truth. We find that agents are more likely to revise when their answers are farther from the correct solution, and that revisions of incorrect answers exhibit a systematic contraction toward the ground truth, even when the final answer remains incorrect. Conversely, correct answers can be overturned by social pressure, particularly when wrong peers are near-correct. Together, these results show that multi-agent interaction induces structured, distance-aware movements in solution space that are invisible under binary correctness, clarifying when social reasoning leads to improvement, stability, or gradual regression in solution quality.

## 1. Introduction

Recent work on large language models (LLMs) has increasingly explored settings in which multiple agents interact, exchange intermediate answers, or revise their responses in light of others (Du et al., 2023; Chan et al., 2023; Yang et al., 2024; Wu et al., 2024). Such collective or social reasoning paradigms have been studied across a range of tasks (Park et al., 2023; Qian et al., 2024; Gottweis et al., 2025),

often reporting improvements or degradations in aggregate performance compared to independent inference or majority voting baselines (Cemri et al., 2025; Zhang et al., 2025a; Choi et al., 2025; Kaesberg et al., 2025). While these results provide useful high-level benchmarks, they leave open a more fundamental question: beyond a simple flip between correct and incorrect, *how does **social interaction reshape an agent's answer** within the space of possible solutions*?

Most existing analyses characterize agent behavior using coarse-grained, categorical indicators such as correctness, answer change rates, or final consensus (Chan et al., 2023; Kaesberg et al., 2025; Zhang et al., 2025a). From this perspective, an agent is typically described as either changed or unchanged, persuaded or not, correct or incorrect, and therefore as succeeding or failing. However, these binary labels obscure potentially rich internal dynamics. Two agents that are both incorrect may differ substantially in how close their answers are to the ground truth, just as two revised answers may represent qualitatively different epistemic moves even if both remain wrong. As a result, aggregate accuracy alone cannot reveal the micro-level dynamics by which collective reasoning succeeds, fails, or stagnates.

Importantly, this limitation is not unique to LLM agents. Decades of research in cognitive science, and social psychology show that human belief updating is graded rather than binary: individuals maintain internal confidence estimates, and become more susceptible to social signals precisely when they perceive their own answers as uncertain or incorrect, while high-confidence beliefs resist social pressure (Asch, 1956; Sporer et al., 1995; Yaniv, 2004), and that being nearly correct can induce qualitatively different revision dynamics than being clearly wrong (Bahrami et al., 2010; Lorenz et al., 2011; Koriat, 2012). If social interaction reshapes beliefs in a distance-sensitive manner in humans, then analyzing collective reasoning in LLM agents solely through binary notions of correctness or answer change risks missing the core mechanisms by which social influence shapes reasoning agents.

In this work, we argue that understanding collective reasoning requires a shift in focus from outcome-level correctness to distance-based microdynamics in the solution space, using the ConceptARC dataset (Moskvichev et al., 2023), a two-dimensional, grid-based reasoning benchmark orga-

[1]Santa Fe Institute, Santa Fe, NM, USA. Correspondence to: Seungwoong Ha <seungwoong.ha@santafe.edu>.

*Proceedings of the 43rd International Conference on Machine Learning*, Seoul, South Korea. PMLR 306, 2026. Copyright 2026 by the author(s).

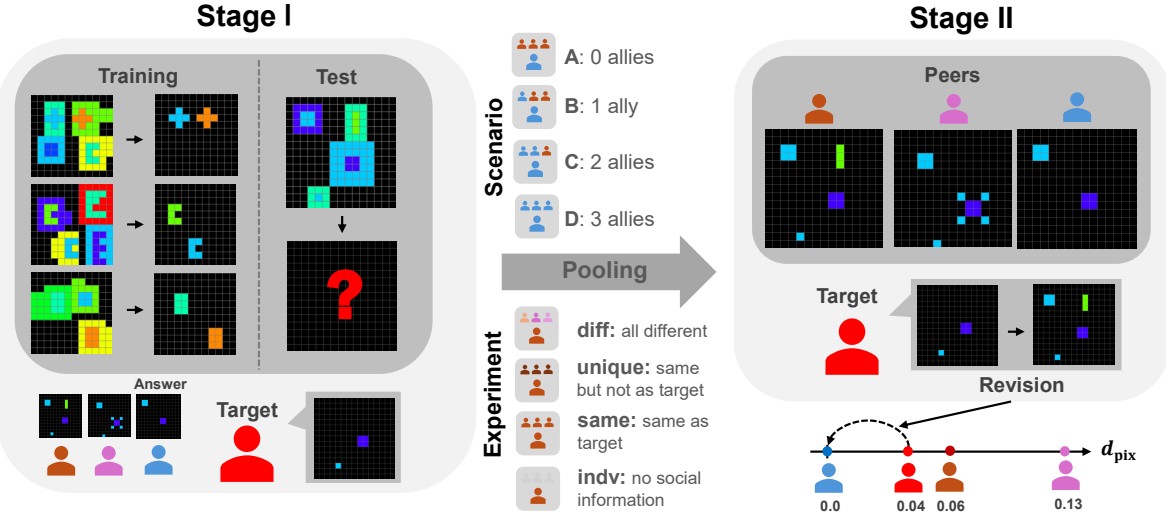

*Figure 1.* Conceptual diagram for MAD settings for a 2D grid rule-inducing reasoning task (ConceptARC). In Stage I, we provide three to four training inputs and outputs, along with one test input, and ask LLM agents to find a common rule to predict the test input's output. We gathered 20 responses for each LLM model, which becomes a baseline sampling pool for Stage II. Depending on the scenario and experiment type, we select a target model, ask peers for $K = 3$ answers, and then ask them to answer again. Finally, the revised answers from the target models are collected, and we analyze the differences between answers using the pixel-wise Hamming distance between the models' responses and the ground-truth labels.

nized around simple semantic concepts. Beyond posing a challenging abstraction task, ConceptARC offers a crucial methodological advantage: answers are structured grids that admit natural distance measures, such as pixel-wise or object-level discrepancies. This allows us to quantify not only whether an answer is correct but also how incorrect it is, and to track how agents move through the solution space under social interaction. By treating an agent's answer as a structured object embedded in a measurable space, our framework allows revisions to be analyzed as continuous movements rather than discrete flips between right and wrong.

Examining revisions in solution space, we show that agents' revision behavior depends systematically on their distance from the ground truth across different models and debate settings, patterns that remain invisible under binary evaluation schemes. By foregrounding distance-aware microdynamics, our approach complements existing correctness-based analyses and provides a more detailed account of how collective reasoning unfolds beneath aggregate performance metrics.

## 2. Related works

**LLM social influence**. Recent studies on MAD have consistently reported that LLMs are affected by social pressure (Cho et al., 2025; Mehdizadeh & Hilbert, 2025), fall into sycophancy (Wynn et al.), or even reach consensus on worse solutions after debate (Chan et al., 2023; Zhang et al., 2025a). Despite these insights, prior work largely treats

all revisions as equivalent events, without distinguishing whether socially induced changes move answers closer to or farther from the underlying solution.

**Fine-grained evaluation for LLM reasoning**. There are diverse lines of work that aim to go beyond simple binary or categorical judgments in LLM reasoning evaluation, seeking to achieve better performance by incorporating finer-grained or continuous evaluation of the model's correct answers and intermediate steps. Verifier guided tree search (VGTS) (Yang et al., 2022), output reward model (ORM) (Luo et al., 2024), and process reward model (PRM) in mathematical reasoning (Zhang et al., 2025b) all use a concept of continuous score to evaluate the quality of LLM reasoning and improve overall accuracy. Also, several studies on MAD have attempted to assess its quality using various metrics, such as average verifier scores (Wang et al., 2024), pairwise judge preferences (Liang et al., 2024), self-reported confidence (Du et al., 2023), and answer similarity (Lewkowycz et al., 2022). Nevertheless, most of these require training an additional model, depending on the presence of external evaluators, or focus on a dataset where there is no natural metric between candidate solutions.

**ConceptARC benchmark** Previous studies on ARC (Chollet, 2019) and ConceptARC (Moskvichev et al., 2023) mainly focused on improving performance and rule assessment (Chollet et al., 2024; Li et al., 2024; Beger et al., 2025), rarely as a testbed for multi-agent collective reasoning and debate (Bikov et al., 2025). In this study, we

leverage the fact that ConceptARC produces structured two-dimensional grid outputs, which naturally induce an image-based distance space among candidate solutions and enable fine-grained comparisons beyond binary correctness.

## 3. Methods

### 3.1. LLM settings

We study how an LLM revises its grid prediction after observing other solvers' answers on the full ConceptARC suite (480 tasks; 16 categories × 10 types × 3 instances). We verify our results by employing two reasoning LLM models: DeepSeek-V3.2 (non-thinking mode, which still performs reasoning) and GPT-5 mini (effort: low), with temperature set to $1.0$ when exposed (otherwise, the API default). Hereafter, we will refer to the two models as DeepSeek-chat and GPT-5 mini, respectively. The experiment consists of two parts: Stage I, which collects responses from individual models, and Stage II, which constructs scenarios based on the collected responses to simulate MAD (Fig. 1).

In Stage I, for each model, we perform a zero-shot prediction task on ConceptARC by providing three to four training examples of input/output grids and directly asking for a common rule to generate the prediction for the single test input. (see Appendix A for the exact prompts). In this study, we focus on the communication between agents solely via their answer grids. To create a baseline pool of possible answers from each LLM, we first collect a baseline pool by running 20 independent generations per task, retaining only valid parsed grids, and aggregating all valid samples.

In Stage II, Social trials are then formed by selecting a target baseline agent with its initial answer, sampling $K = 3$ peer answers from answer pools for the same task according to a scenario and an experiment type, and querying the target model once more with a 3-message history (original task prompt, target initial answer grid, and a re-prompt with sampled peer answers). For more details and exact prompts used for simulation, see Appendix A.

### 3.2. Experiment types and scenarios

Scenario letters, ranging from A to D, indicate how many of the provided peers match the target model's answer type (correct or incorrect). More specifically, when the target model's initial answer is wrong: A: C3, B: W1C2, C: W2C1, D: W3, and when the target is correct: A: W3, B: W2C1, C: W1C2, D: C3 are provided to the target model, where $Ck$ denotes $k$ correct peers and $Wk$ denotes $k$ wrong peers.

Within a fixed scenario (i.e., a fixed number of correct/wrong peers), the experiment type specifies how the target model's answers are set and how wrong peers are instantiated relative to the target. For the cases where the target model answer is

correct, we have `uniqueCorrect` (UC): wrong peers repeat a single wrong answer, `diffCorrect` (DC): wrong peers are different from each other, and `indvWrong` (IW): no peers provided. When the target model answer is wrong, we have `sameWrong` (SW): all wrong peers are identical to the target, `uniqueWrong` (UW): all wrong peers are identical, different from the target, `diffWrong` (DW): wrong peers are all different from each other, and `indvCorrect` (IC): no peers provided.

These settings are designed to allow separate examination of various conditions: the target's previous answer type (correct or incorrect), the number of peers sharing the same type, the social pressure from identical answers, and the possibility of self-reflection. Note that some of the combinations of scenarios and experiment types are identical to each other (see Appendix A.4.3). All code for LLM simulations and figure generation is available at https://github.com/nokpil/socialLLM.

## 4. Categorical analysis of revised answers

To begin, we analyze the revised answers from target agents using a more conventional approach: measuring whether the target model changes its answer grid and which categories the answer falls into. The overall pass@1 accuracy for the entire suite was $37.16\%$ (DeepSeek-chat) and $56.82\%$ (GPT-5 mini), making GPT-5 mini the better-performing model. Invalid responses (less than $0.4\%$) are removed from the analysis. See Table 1 and 1 for exact sample counts in Stage II for each scenario after removing invalid responses.

### 4.1. Correct target case

For targets with the correct initial answers (Fig. 2a, d), a larger number of wrong peers increases the likelihood that a correct answer is overturned: scenario A (W3) is consistently more destabilizing than scenarios with fewer wrong peers. Thus, even correct solutions are vulnerable when the social signal is uniformly opposed. Interestingly, the probability of stable revision (kept their correct initial answers) exceeds the ratio of correct answers among their social group: that is, even if only two out of four members in a social group have the correct answer, the probability that each correct responder will maintain their answer far exceeds $50\%$. This is a significant difference compared to the wrong-target case, which will be shown shortly.

Beyond the number of wrong peers, their internal structure plays a decisive role. When all wrong peers present the same incorrect answer (`uniqueCorrect`), the target is more likely to abandon its correct solution than when they are diverse (`diffCorrect`). This suggests that identical answers serve as a strong conformity signal, exerting disproportionate social pressure compared with heterogeneous

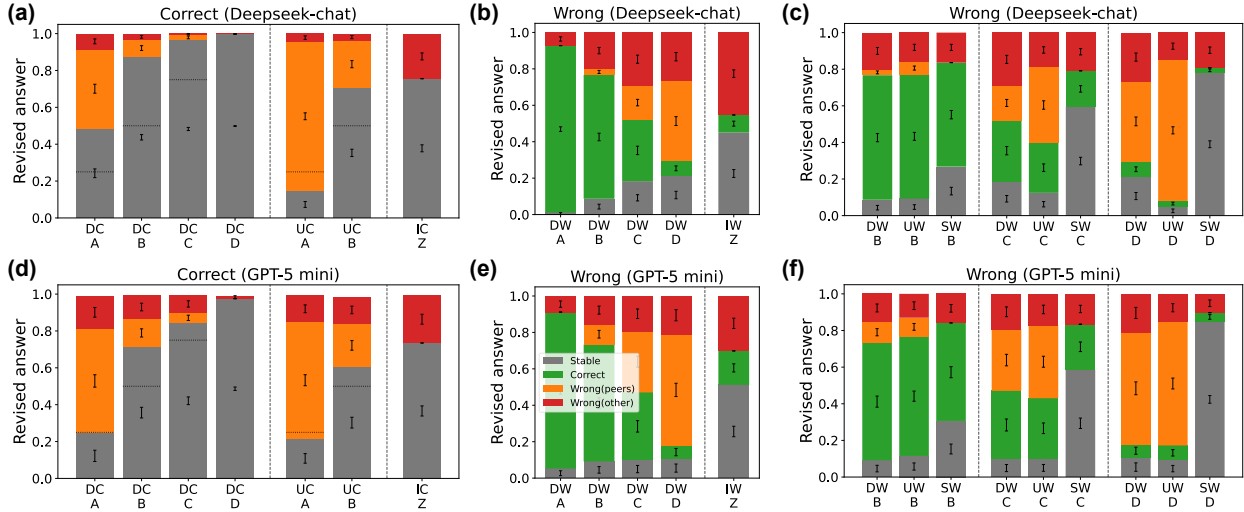

*Figure 2.* Revised answer distribution from (a-c) DeepSeek-chat and (d-f) GPT-5-mini. Correct target cases are depicted in (a, d), while the wrong target cases are divided into (b, e) and (c, f) for comparison between different experiment types. Black dotted lines on bars in (a, d) indicate the ratio of correct agents among the social group for each scenario. Error bars indicate 95% CI by multi-categorical bootstrap methods with 1,000 samples.

disagreement. This effect is consistent with findings on social conformity in humans (Asch, 1956; Lorenz et al., 2011) and LLM (Chan et al., 2023; Kaesberg et al., 2025).

Finally, we compare social influence against self-reflection by examining individual reprompting conditions (`indvCorrect`). In the absence of any peer answers, both models show high stability, with more than 70% of correct targets retaining their original answers after reprompting. When combined with a high retention rate in other social scenarios, these results demonstrate that correct answers are generally robust under social pressure and self-reflection, but not completely invincible in the face of strong, homogeneous social pressure.

### 4.2. Wrong target case

We next turn to cases in which the target agent's initial answer is incorrect (Figs. 2b-c, e-f), and examine how social exposure facilitates or inhibits correction. As expected, the presence of correct peers strongly promotes revision: scenarios with more correct peers (e.g., A, C3) lead to substantially higher rates of answer change than scenarios with more incorrect peers. This confirms that correct social signals can effectively overturn an initially incorrect belief.

However, the effect of wrong peers depends critically on their relation to the target's own answer. When wrong peers differ from the target's answer (`uniqueWrong` or `diffWrong`), the target model surrenders its answer quite easily, showing less than 10% of retention rate. But, when all wrong peers are identical to the target's wrong answer

(`sameWrong`), the target exhibits markedly increased stability that is nearly comparable to the correct target case, even in the presence of correct peers. This configuration effectively reinforces the target's belief, acting as a form of confederate support that suppresses revision (Asch, 1956).

Wrong targets are also less resilient under self-reprompting. In individual conditions without peers (`indvWrong`), 50% or less of wrong answers remain unchanged after reconsideration, indicating that incorrect solutions are intrinsically less stable even in the absence of social pressure. Together, these results show that social interaction can either promote correction or entrench error, depending not only on the number of correct peers but also on whether wrong answers align with the target's existing belief.

## 5. Distance dependency of answer revision

The analyses in the previous section characterize revision behavior using categorical outcomes, such as whether an answer changes and whether it is correct after revision. While informative, this perspective treats all incorrect answers as equivalent. In practice, however, wrong answers vary widely in how close they are to the ground truth: some differ by only a few pixels, while others are qualitatively far from any plausible solution (see Fig. 3(a) for examples). Collapsing these heterogeneous states into a single "wrong" category obscures potentially systematic patterns in how agents revise their beliefs.

This limitation is particularly evident when the problem contains numerous plausible-sounding wrong answers and

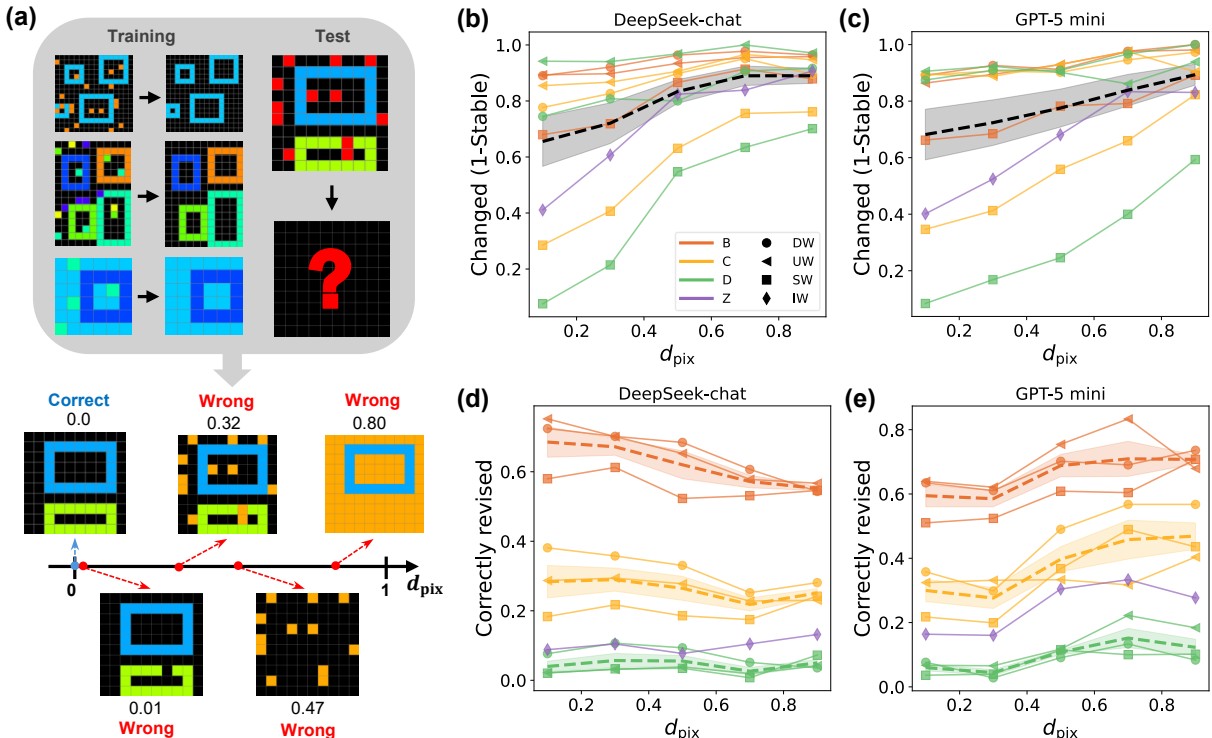

*Figure 3.* Distance-dependent revision behavior for wrong target cases. (a) Example of multiple different wrong answers from the example problem in ConceptARC (Copy4-3). (b-c) Probability of changed answers (equals to 1 minus the ratio of stable answers) among the wrong target cases for (b) DeepSeek-chat and (c) GPT-5 mini, by their target's pixel distances to the answer. (d-e) Probability of correctly revising answers from the wrong target cases for (d) DeepSeek-chat and (e) GPT-5 mini. Thin colored lines indicate individual experiment and scenario conditions, where the thick dotted line represents the average across (b-c) all conditions and (d-e) each scenario, with the shaded region indicating ±1 SEM.

features a complex, multi-step reasoning structure. An agent may resist revision because its answer is internally coherent and nearly correct, or revise readily because its answer is evidently inconsistent on the second thought, even though both are labeled as incorrect. Similarly, revisions that remain wrong after social interaction may nonetheless represent meaningful epistemic progress by moving closer to the true solution. This is under a purely categorical evaluation; such improvements are indistinguishable from arbitrary or regressive changes.

ConceptARC provides a natural testbed for addressing this issue, as candidate answers are structured in two-dimensional grids that admit quantitative distance measures. This allows us to treat each answer as a point in a structured solution space and to analyze revisions as movements within that space rather than as binary flips. In this study, we quantify answer quality using the pixel-wise normalized Hamming distance, $d_{\text{pix}}(Y, \hat{Y}) = 1 - \frac{1}{HW} \sum_{r=1}^{H} \sum_{c=1}^{W} \mathbb{I}\left[ Y = \hat{Y} \right]$, which is simply one minus the fraction of equal cells. Here, $Y$ is the answer from the LLM agent (after the grid regularization, see Appendix A.5)

and $\hat{Y}$ is the ground-truth with height and width of $(H, W)$.

This intuitive pixel-wise Hamming distance serves as a good proxy for the difference between the answer and the ground-truth label (and becomes $0$ when the answer matches the label) and directly indicates output fidelity, but there may be cases in which the simple pixel-wise distance cannot fully capture the meaningful semantic distances between different answers. For instance, Hamming distance might fail to capture the appropriate semantic distance when the problem involves a shift or rotation (as in the MoveToBoundary problem in ConceptArc) or when the output format is exotic (e.g., the answer is not formatted with the image but in a more conceptual form). To supplement this, we report two auxiliary distance metrics for structural consistency (see Appendix B): a color-aware earth mover's distance and an object-level edit distance based on connected components and Hungarian matching, which are inspired by methodologies from image comparison tasks (Kuhn, 1955; Munkres, 1957; Blaschke, 2010; Ma & Latecki, 2011) and complement potential semantic gaps in the Hamming distance. All results for these auxiliary distance metrics are in the Ap-

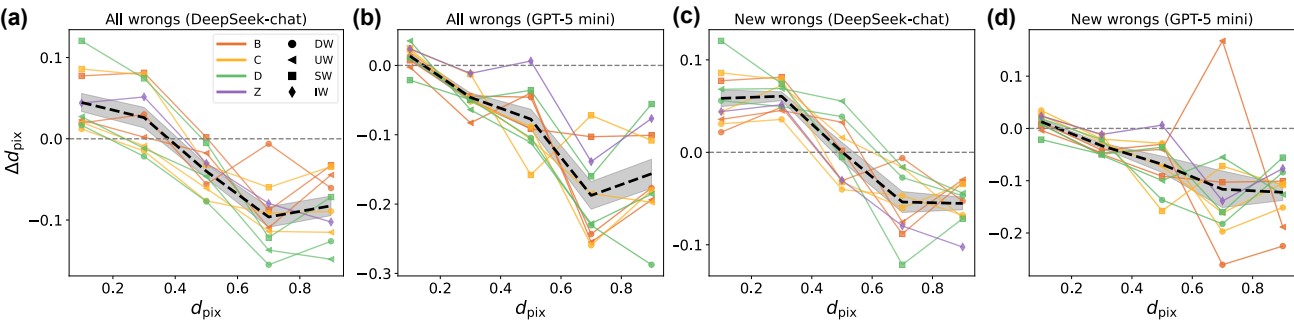

*Figure 4.* Distance contraction after revision for wrong target cases. (a-d) Distance change after revision ($\Delta d_{\text{pix}} = d_{\text{pix}}^{\text{after}} - d_{\text{pix}}^{\text{before}}$) from the wrong target cases, aggregated for (a-b) all wrong answers that are different from the target's initial answer, and (c-d) newly generated wrong answers different from the provided peer answers. Results are from (a, c) DeepSeek-chat and (b, d) GPT-5 mini. Thin colored lines indicate individual experiment and scenario conditions, where the thick black line represents the average, with the shaded region indicating $\pm 1$ SEM.

pendix (see Appendix C) and mostly align with the findings below.

## 5.1. Initial distance dependency of revised wrong answers

We begin by examining how the initial distance between a target's answer and the ground truth affects its likelihood of revision. Motivated by analogies to confidence–accuracy relationships in human decision-making (Sporer et al., 1995), we hypothesize that target agents whose answers are farther from the correct solution are more likely to revise when exposed to social information or self-reprompting. Intuitively, answers that are far from the ground truth are less likely to be internally consistent, making the agent less confident in its solution and more likely to recognize obvious inconsistencies when given another opportunity to reconsider.

### 5.1.1. PROBABILITY OF CHANGING ANSWER

Figures 3 (b,c) show the probability that a wrong target changes its answer as a function of the initial distance $d_{\text{pix}}$. Across both models, we observe a clear monotonic trend: targets with answers farther from the correct solution are substantially more likely to revise. Even without any explicit signal about correctness, agents whose initial answers have large $d_{\text{pix}}$ overturn their beliefs much more frequently than those whose answers are closer to the label. This pattern is robust across social scenarios and is also present in individual re-prompting conditions (indvWrong, purple), suggesting that distance captures an intrinsic aspect of answer stability.

Notably, this distance dependence persists even when social pressure favors stability. In the sameWrong condition, where all wrong peers exactly match the target's wrong answer, targets are nominally supported by a unanimous majority. Nevertheless, when $d_{\text{pix}}$ is large, targets still revise their answers more than half of the time, even in the extreme case of scenario D. This indicates that strong social agreement is insufficient to fully stabilize answers that are internally inconsistent or far from the ground truth.

### 5.1.2. PROBABILITY OF CORRECTLY CHANGING ANSWERS

We next ask whether these revisions improve correctness by assessing the probability of correct answers after the social trial. Again, intuitively, Figs. 3(d,e) reports the probability that a revised answer becomes correct as a function of the initial distance. Here, the two models exhibit qualitatively different behaviors. For DeepSeek-chat, a high initial distance is mostly associated with lower correction rates (especially in scenario B (orange), where only one correct peer is provided). In contrast, GPT-5 mini shows a pronounced positive trend: targets with larger initial distances are not only more likely to revise, but also more likely to revise correctly. For this better-performing model, higher revision rates at large $d_{\text{pix}}$ correspond to genuine epistemic improvement rather than random exploration.

Taken together, these results show that the initial distance to the ground truth is a strong predictor of both the likelihood of revision and, for sufficiently capable models, successful correction. At the same time, many revisions at a large distance remain incorrect when the target was mainly exposed to incorrect answers (in scenarios like C (yellow) and D (green)). This raises a natural follow-up question: when a revised answer is still wrong, does it nonetheless move closer to the correct solution? In other words, *does the again-wrong LLM agent still benefit from social trial?*

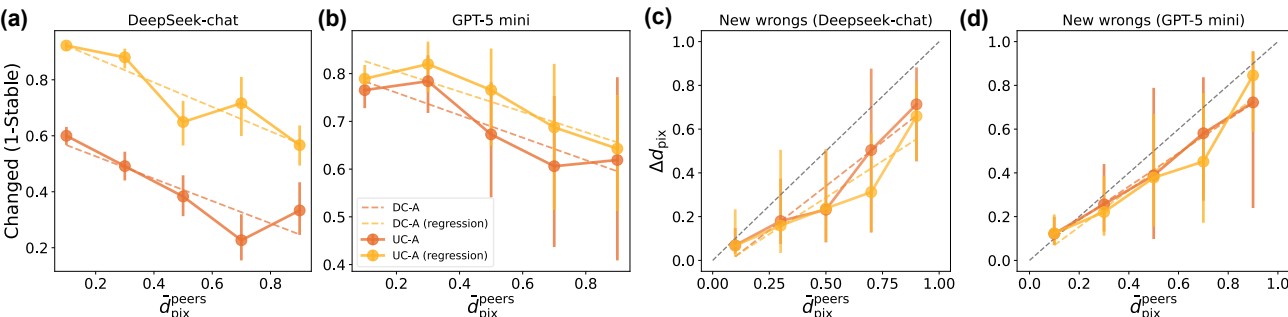

*Figure 5.* Distance-aware social influence for correct target cases under wrong peers (scenario A). (a-b) Probability of changing answer as a function of the mean peer distance $\bar{d}_{\text{pix}}^{\text{peers}}$ for all incorrect answers that differ from the target's initial answer. (c-d) Distance of newly generated wrong answers after revision. Results are from (a,c) DeepSeek-chat and (b,d) GPT-5 mini. Error bars indicate $95\%$ Wilson score interval.

## 5.2. Distance contraction after revision of wrong answers

We address the previously raised question by analyzing how the distance to the ground truth changes after revision, focusing on cases where the revised answer differs from the original yet remains incorrect. Figure 4 shows the difference between the original answer and the revised answer $\Delta d_{\text{pix}} = d_{\text{pix}}^{\text{after}} - d_{\text{pix}}^{\text{before}}$ as a function of the initial distance $d_{\text{pix}}^{\text{before}}$ for wrong-target cases.

### 5.2.1. COPIED WRONG ANSWERS FROM PEERS

We first check the average $\Delta d_{\text{pix}}$ of the entire wrong answers from the target (Figs. 4 (a,b)). Across both models, we observe a clear negative trend between the initial distance and the change in post-revision distance. Targets with high $d_{\text{pix}}^{\text{before}}$ tend to revise toward smaller distances, even when the revised answers are still wrong. Conversely, targets with low $d_{\text{pix}}^{\text{before}}$ exhibit smaller average changes. This contraction effect indicates that revisions are not random perturbations in solution space, but are systematically biased toward the ground truth.

Importantly, all revisions analyzed here would be classified as failures under a binary correctness criterion, despite substantial improvements in distance. This highlights a key limitation of categorical evaluation: it cannot distinguish between wrong answers that move closer to the solution and those that drift arbitrarily or deteriorate.

### 5.2.2. NEWLY GENERATED WRONG ANSWERS

One potential concern is that this contraction could arise trivially from copying peer answers. Since peers are uniformly sampled from a pool of wrong answers, targets with a large initial distance may encounter peers closer to the label, leading to a mechanical reduction in distance (and vice versa). To address this, we separately analyze "new" wrong

answers that do not exactly match any of the provided peer answers (Figs. 4(c,d)). The negative correlation persists in this subset as well, confirming that contraction reflects genuine revision rather than direct imitation.

Here, two models exhibit notable qualitative differences. For DeekSeek, targets with small initial distances (less than 0.5) sometimes show mild deterioration (an increase in distance) after revision, suggesting that near-correct answers can still be tempted and destabilized by social influence. In contrast, the better-performing GPT-5 mini shows little to no deterioration under these conditions when the initial distance is small. Revised answers tend to remain close to the ground truth, while large-distance answers consistently improve. This asymmetry suggests that higher-capability models are better able to preserve near-correct internal structure while still correcting gross errors.

Overall, these results demonstrate that even unsuccessful revisions are often epistemically meaningful and beneficial in the long run. Social interaction induces a contraction of error toward the ground truth, particularly for answers that are initially far from it. This distance-aware perspective reveals systematic learning dynamics that are invisible under binary correctness or categorical analysis and clarifies how multi-agent interaction can promote gradual improvement even in the absence of immediate success.

## 5.3. Distance effects for correct targets under wrong social signals

The distance-dependent patterns identified above are not confined to wrong-target cases. We find that similar distance-aware dynamics also emerge when the target's initial answer is correct, but it is exposed to wrong peers. In such settings, social influence does not act uniformly: the effect depends systematically on how close the incorrect answers provided are to the correct solution.

To isolate this effect, we focus on scenario A, in which all three peers provide incorrect answers. This configuration represents the strongest social pressure for a correct target to abandon its own answer. Since the target's initial distance to the ground truth is always zero in these cases, we instead characterize the social signal by the mean peer distance $\bar{d}_{\text{pix}}^{\text{peers}}$, defined as the average pixel-wise distance of the three wrong peer answers to the label grid.

Figures 5(a,b) show the probability that a correct target changes its answer as a function of $\bar{d}_{\text{pix}}^{\text{peers}}$. Across both models, we observe a clear trend: correct targets are more likely to overturn their answers when the wrong peers are closer to the ground truth. Near-correct wrong answers exert substantially stronger influence than distant, clearly implausible ones, indicating that social pressure is mediated by the apparent plausibility of the alternatives rather than by their correctness alone.

We next examine the nature of the resulting errors. Figures 5(c,d) plot the distance of newly generated wrong answers from correct targets after revision. Because $d_{\text{pix}}^{\text{before}} = 0$ in these cases, the distance change reduces to $\Delta d_{\text{pix}} = d_{\text{pix}}^{\text{after}}$. In both models, the average revised distance lies below the identity line, showing that newly created wrong answers typically remain close to the mean peer distance. Thus, when correct targets are overturned, the resulting deterioration is gradual rather than catastrophic, with revised answers gravitating toward nearby incorrect alternatives instead of drifting arbitrarily.

Taken together, these results show that distance-aware dynamics operate symmetrically across correct and wrong targets. Social influence is strongest when alternative answers are close to the truth, and revisions tend to be local in solution space. This further reinforces the view that multi-agent interaction induces structured, distance-sensitive movements.

For the additional analyses, we showed that (1) the results are reproducible with a Hamming distance-faithful subset of the ConceptARC, (2) the contraction cannot be fully explained by a simple soft imitation of peers, (3) the main messages of this work holds in a multi-step interaction as well, (4) backup evidence for systemic difference in Fig. 3(d, e) with another model (Gemini-2.5-flash-lite), and (5) revision rates changes with the different prompt but the overall tendency persists (check the Appendix D for more details).

## 6. Discussion

The results in Sections 4 and 5 collectively show that multi-agent debate reshapes LLM behavior through structured, distance-aware dynamics rather than through simple categorical flips between correctness and error. Across wrong and correct target cases alike, the likelihood, direction, and consequence of revision depend systematically on how close an answer is to the ground truth or to plausible alternatives. Agents that are far from the correct solution are more likely to revise and, in many cases, move closer to it (Figs. 3 and 4), while agents that are initially correct are most vulnerable to social influence when confronted with near-correct wrong answers (Fig 5). Crucially, even revisions that remain incorrect are not arbitrary: they tend to be local movements in solution space, exhibiting contraction toward the true answer or toward nearby incorrect alternatives rather than catastrophic drift.

Together, these strongly consistent findings indicate that social interaction induces graded, geometry-aware belief updates, revealing epistemic microdynamics that are invisible under binary, correctness-based evaluation and that align with collective problem-solving in human groups (L Griffiths et al., 2008; Koriat, 2012). Notably, it seems worth examining why LLMs behave as if they possess metacognition about their own answers—even when they have never been asked to express confidence separately and have received no external signals about the correctness of their responses—as their revision dynamics suggest. In fact, assessing the reliability of LLM responses is a domain in which extensive research is underway to improve performance, fidelity, and security. Various methodologies are being widely studied, such as asking the LLM to state its confidence level for its answer (Tian et al., 2023; Xiong et al., 2024), training it to provide confidence levels during training (Lin et al., 2022; Kadavath et al., 2022), or indirectly measuring answer reliability using final-token probabilities (Plaut et al., 2024) or the variance of answers across multiple iterations (Manakul et al., 2023; Becker & Soatto, 2024). By applying these methodologies to this study—such as closely examining the reasoning behind the process by which LLM agents revise their answers after receiving peer answers, requiring them to provide confidence scores alongside their answers, or examining the model's internal workings—and analyzing the correlation between the confidence scores of correct answers and the revision rate, we can gain a deeper understanding of how closely the social revision dynamics of LLMs resemble those of humans.

Understanding distance-aware dynamics is important in MAD and AI organizational design as well, because they can lead to outcomes that differ qualitatively from those predicted by conventional MAD evaluations. In particular, there exist regimes in which raw accuracy remains low (so that majority voting cannot recover the correct answer), and even a single round of interaction has little chance of turning an initially wrong answer into a correct one, yet multi-agent discussion consistently reduces the average distance to the ground truth. In these cases, although early-stage majority voting is ineffective, iteratively discussing multiple steps

and applying the majority rule only then can yield solutions that are unreachable by any single-shot aggregation. This distinction becomes especially relevant in tasks where exact correctness is rare and partially correct solutions remain meaningful, such as difficult mathematical reasoning or open-ended problem-solving.

Beyond their theoretical interest, the distance-aware dynamics identified in this work have concrete implications for the design and evaluation of multi-agent systems. First, our finding that even incorrect revisions systematically contract toward the ground truth suggests that iterative debate pipelines can accumulate epistemic progress across rounds, even when single-round accuracy gains appear negligible: a regime in which conventional majority voting would prematurely declare failure. This motivates evaluation protocols that continuously track solution quality rather than measure only final correctness. Second, and perhaps more critically, the asymmetric vulnerability of correct agents reveals a security-relevant insight: socially induced errors are most likely to arise not from clearly wrong peer answers, but from near-correct ones that appear locally plausible. This implies that adversarial manipulation of LLM agents does not require fabricating obviously false information; rather, *strategically crafted* near-correct distractors constitute a more effective and harder-to-detect attack vector. As LLM-based multi-agent systems are increasingly deployed in high-stakes settings, understanding and mitigating this vulnerability represents an important direction for future work.

Motivated by this observation, a natural next step is to move beyond one-step revision and examine whether such distance-aware dynamics persist across multiple rounds of interaction. In this work, we deliberately restricted our attention to single-step updates, as independently testing diverse social configurations already yields exponential growth in the number of possible trajectories once fine-grained distance metrics are introduced. Nevertheless, if the contraction and locality effects identified here remain stable over repeated rounds, this would provide much stronger evidence that distance-aware social dynamics constitute a genuine learning mechanism rather than a transient artifact of one-shot exposure. Multi-round analysis would also enable the study of long-term convergence, oscillation, or drift in solution space (DeGroot, 1974). With sufficient experimental data, they open the door to a probabilistic modeling perspective of long-run collective behavior in which answer updates are described by transition probabilities conditioned on epistemic distance (Deffuant et al., 2000; Acemoglu & Ozdaglar, 2011). Through additional experiments, we confirmed that these effects are indeed maintained, albeit to a limited extent, even in multi-step social interactions (see Appendix D.3).

Another promising direction builds directly on ConceptARC's rule-inducing nature. Rather than exchanging final grids alone, agents could be asked to explicitly communicate the rules they inferred, after which social influence could be studied in a space of rules rather than the solution itself. Because ConceptARC is designed with simple, concise rule descriptions, this setting is particularly well-suited for defining and measuring distances between rules, for example, via symbolic structure or embedding-based similarity (Lake et al., 2017). Such a framework would bring the setup closer to genuine debate, while remaining more tractable than reasoning tasks that require comparing full chains of thought or intermediate derivations.

Finally, it would be very interesting to investigate whether the distance-based, fine-grained social revision dynamics identified in this study also exist in other domains—such as math problem-solving, programming, and multidimensional optimization—and, if so, what form they take. As mentioned in the introduction, while many previous studies have attempted to assign fine-grained scores to answers, these were generally introduced solely to improve the performance of the AI population as a whole, and research on how these scores change with each social interaction remains scarce. Furthermore, there is very little research examining whether such distance metrics align with the semantic distance (from the correct answer) as assessed by actual humans. As with the aforementioned rule in ConceptArc, mapping these dynamics into the embedding space or effectively training multidimensional metrics to detect similar continuous patterns across domains will be essential for understanding the problem-solving dynamics of AI systems, which are expected to be applied across diverse fields in the future.

## 7. Conclusion and Outlook

In conclusion, this work suggests that many debates surrounding the effectiveness or failure of multi-agent debate stem from overly coarse evaluation criteria. By revealing distance-aware, geometry-sensitive revision dynamics, we show that social interaction can induce structured learning signals even when accuracy improvements are small or absent. These insights have implications beyond ConceptARC, informing the design, evaluation, and comparison of societies of AI agents for collaborative problem-solving. More broadly, these insights point toward a view in which progress in multi-agent reasoning will depend not only on better agents but also on richer notions of what it means to move closer to understanding.

## Impact Statement

This paper presents work aimed at advancing the field of Machine Learning. There are many potential societal consequences of our work, none of which we feel must be specifically highlighted here.

## Acknowledgement

SW acknowledges generous support from the Emergent Political Economies grant, provided by the Omidyar Network to the Santa Fe Institute. MM gratefully acknowledges support from the grant "Building Diverse Intelligences through Compositionality and Mechanism Design" from the Templeton World Charity Foundation, https://doi.org/10.54224/33804 and https://doi.org/10.54224/20650k, and from the BANYAN project at the Sandia National Laboratories, subcontract 2673699. We also thank

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

# A. Methods

## A.1. Task Suite

We evaluate on the *full* ConceptARC suite: 16 categories × 10 problem types per category × 3 instances per type, totaling 480 tasks. Each task specifies training input-output examples and test inputs. The goal is to output the correct test grid as a 2D array of integer color indices, ranging from 0 to 9.

## A.2. Models and Generation Settings

We consider two LLM solvers: DeepSeek-V3.2 (non-thinking mode; chat) and GPT-5 mini (low effort setting). All runs are executed by calling the official API with the temperature fixed to 1.0 in our pipeline. For GPT-5 mini, the official API used in our code does not expose the temperature parameter; thus, it is effectively fixed at the API default (treated as "temperature-fixed").

## A.3. Stage I: Independent Sampling and the Baseline Pool

To obtain an empirical distribution of each model's outputs, we run an *independent* baseline prompting procedure 20 times per task (total baseline calls: 480 × 20). All baseline samples are parsed into grids, and invalid parses are removed. The remaining valid samples are aggregated, forming the source of resampled (incorrect) peer answers in Stage II.

Stage I uses a single-user message containing the full task description, including formatted training examples and the test input. We provide all grids as concatenated text, such as "6 6 6\n 4 4 4\n 3 2 1". The prompt requires a JSON output containing only the final grid. The template is shown in Fig. 6.

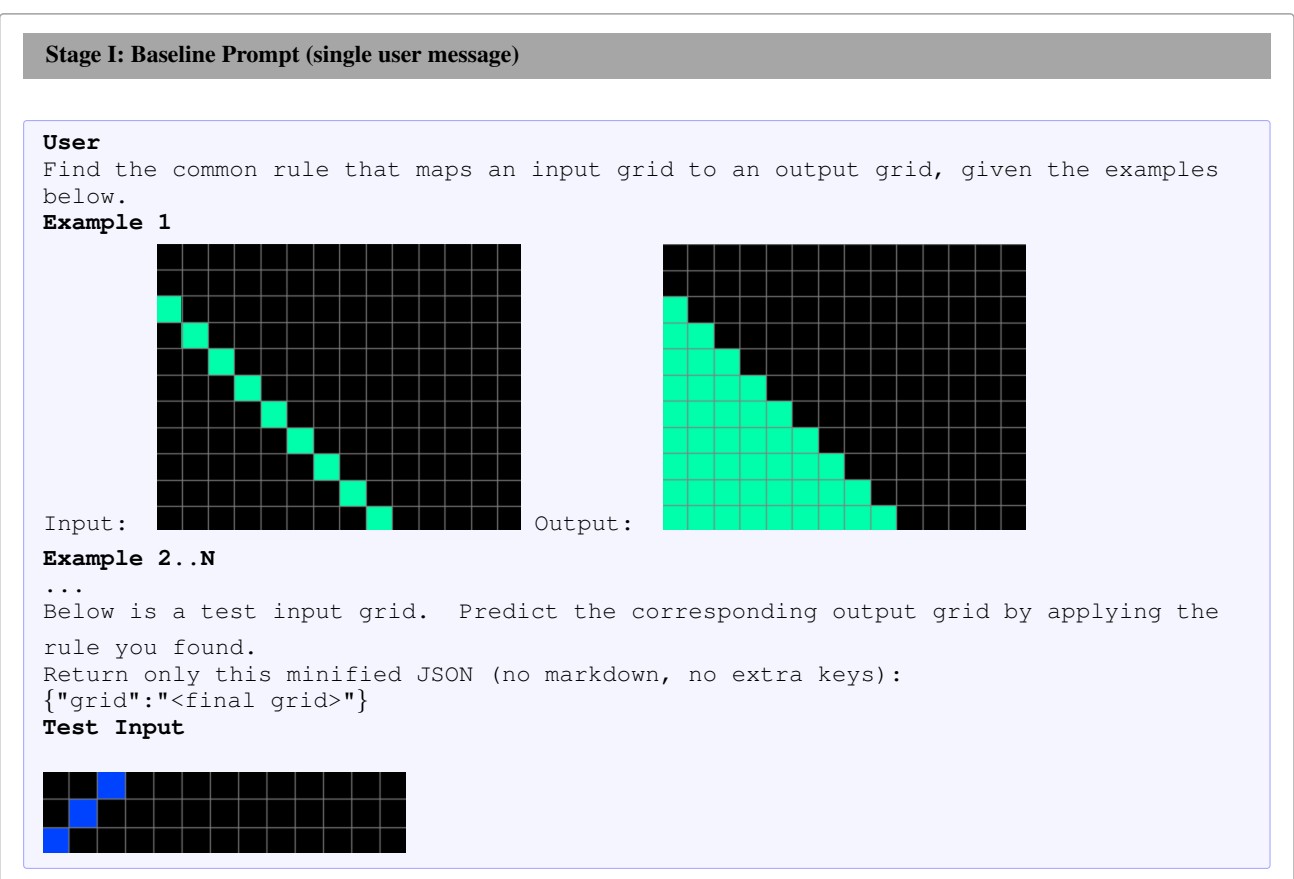

*Figure 6.* Stage I baseline prompt template (single user message). Grids are represented as images for visualization, while the LLM receives them as concatenated text.

**A.4. Stage II: Social Revision by Resampling Peer Answers**

Stage II simulates one-step social influence by presenting a target LLM model (agent) with: (i) the original task prompt, (ii) its own previous answer, and (iii) three peer answers (sampled from stage I) for the same task (Fig. 7. Hereafter, we call the main agent that will be queried again the *target*, and the additional agent providing answers to the target the emph peer. Note that these peers are not newly generated. To reduce computational cost, we resampled and provided baseline attempts as peer solvers to simulate social interaction, enabling controlled scenario construction and efficient verification.

A.4.1. CASE CONSTRUCTION AND TRIAL COUNTS

We define cases for the target model from the baseline pool at stage I. For target-wrong experiments, we enumerate the unique wrong grids produced by a given model across the 10 baseline runs and treat each unique wrong grid as a target model instance. This yields 1952 target-wrong cases for DeepSeek-chat and 1078 for GPT-5 (since it had fewer unique wrong answers) among all 480 ConceptArc suites.

For target-correct experiments (`uniqueCorrect` and `diffCorrect`), we construct cases on the same task-wise support so that the number of target-correct cases matches the target-wrong cases: each target-correct case uses the task's correct grid as the target initial answer while retaining the task's wrong-set metadata for sampling peer wrong answers. Additionally, different from `indvWrong`, where we can simply set all target-wrong cases as target model instances, `indvCorrect` contains exactly one target per task (the unique correct grid), for a total of 480 target-correct cases (since there could be only one correct answer grid to a given problem).

**Repetitions.** For each target case and each scenario (Section A.4.2), we run exactly one social trial (no repeated resampling for the same target–scenario pair). Thus, per model and per experimental type, the number of trials is (#targets) × 4 for A/B/C/D scenarios (with some exceptions which will be explained below), and (#targets) for individual (Z) scenarios.

A.4.2. SCENARIO DEFINITION (A/B/C/D)

Each social trial shows $K = 3$ peer answers. The scenario letter $S \in \{\texttt{A}, \texttt{B}, \texttt{C}, \texttt{D}\}$ controls the number of correct peers versus wrong peers among the three, while the experimental type controls the *structure of the wrong peers* (Section A.4.3).

**target-wrong scenarios.** When the target initial answer is wrong, we define:

$$\texttt{A} : (C3), \quad \texttt{B} : (W1C2), \quad \texttt{C} : (W2C1), \quad \texttt{D} : (W3),$$

where $Ck$ denotes $k$ correct peers (label-matching), and $Wk$ denotes $k$ wrong peers (non-label). Thus, scenarios vary the amount of "correct social signal" opposing the target's wrong answer.

**target-correct scenarios.** When the target initial answer is correct, we use the complementary structure:

$$\texttt{A} : (W3), \quad \texttt{B} : (W2C1), \quad \texttt{C} : (W1C2), \quad \texttt{D} : (C3),$$

so that the scenario letter consistently represents the strength of agreement with the label among peers.

A.4.3. EXPERIMENT TYPES

Within a fixed scenario (i.e., a fixed number of wrong-peer slots $Wk$), the experiment type specifies how the wrong peers are instantiated relative to the target's initial answer (see Fig. 1).

For target-correct conditions, peers in scenarios A/B/C include wrong answers; we use:

- Correct target case:
    - `uniqueCorrect` (UC): wrong peers repeat a single wrong grid.
    - `diffCorrect` (DC): wrong peers are diverse wrong grids (as available).
    - `indvCorrect` (IC): no peers are shown; the target is reprompted once, starting from its prior correct answer.
- Wrong target case:

- `sameWrong` (SW): all wrong peers (if any) are *identical to the target's wrong grid*. For example, under scenario B, the peers are $(W1C2)$ where the single $W$ equals the target wrong.
- `uniqueWrong` (UW): all wrong peers (if any) repeat a *single alternative wrong grid* that is distinct from the target wrong.
- `diffWrong` (DW): wrong peers (if any) are *different from the target and from each other*; when multiple wrong peers are required (scenarios C/D), they are sampled to be mutually different as available (up to three alternatives).
- `indvWrong` (IW): no peer answers are shown; the target is reprompted once to output its best current answer.

In all cases, we sample at most three distinct wrong alternatives per target case to populate the required $Wk$ slots. Some of the combinations are identical in nature: UC-C/D and DC-C/D are the same, since `unique` and `diff` bear no difference when there are only $1/0$ wrong answers provided.

Table 1 lists the number of total and valid samples (after the filtering in stage II) used in each experimental condition, broken down by model (DeepSeek-chat and GPT-4o-mini), experiment type, and scenario. In table 1, the total sample counts for each figure are listed. Note that there is a systematic decrease in the valid response rate in diffWrong ($B > C > D$), suggesting that both models suffer from a more difficult social situation, with no consensus at all. Note that this does not affect the main findings of the study, since all of those 'invalid' responses will be regarded as 'wrong' even if we include them (intuitively, all of those 'invalid' responses would simply increase the 'red' part in Fig.2, which is already increasing as $B < C < D$ in diffWrong). This might be alleviated by adding a strong constraint on JSON-parsable output, as the current version of the prompt is deliberately minimalistic.

*Table 1.* Sample counts per experimental condition. $N_{\text{tot}}$: total number of samples. $N_{\text{valid}}$: number of samples used after removing unparsable outputs. Ratio: $N_{\text{valid}}/N_{\text{tot}}$.

| | | DeepSeek-chat | | GPT-4o-mini | |
|---|---|---|---|---|---|
| **Exp. type** | **Scenario** | $N_{\text{tot}}$ | $N_{\text{valid}}$ (%) | $N_{\text{tot}}$ | $N_{\text{valid}}$ (%) |
| `diffCorrect` (DC) | A | 1617 | 1607 (99.38) | 820 | 818 (99.76) |
| | B | 1868 | 1850 (99.78) | 968 | 968 (100) |
| | C | 1952 | 1952 (100) | 1078 | 1078 (100) |
| | D | 1952 | 1952 (100) | 1078 | 1078 (100) |
| `uniqueCorrect` (UC) | A | 1952 | 1852 (94.88) | 1078 | 1040 (96.47) |
| | B | 1952 | 1939 (99.33) | 1078 | 1076 (99.81) |
| `indvCorrect` (IC) | Z | 480 | 479 (0.99) | 480 | 480 (100) |
| `diffWrong` (DW) | A | 1952 | 1950 (99.90) | 1078 | 1078 (100) |
| | B | 1868 | 1850 (99.52) | 974 | 974 (100) |
| | C | 1726 | 1716 (99.42) | 830 | 827 (99.64) |
| | D | 1558 | 1543 (99.04) | 677 | 672 (99.26) |
| `uniqueWrong` (UW) | B | 1868 | 1856 (99.36) | 974 | 974 (100) |
| | C | 1868 | 1848 (98.93) | 974 | 969 (99.49) |
| | D | 1868 | 1798 (96.25) | 974 | 945 (97.02) |
| `sameWrong` (SW) | B | 1952 | 1948 (99.80) | 1078 | 1078 (100) |
| | C | 1952 | 1942 (99.49) | 1078 | 1073 (99.54) |
| | D | 1952 | 1899 (97.28) | 1078 | 1052 (97.59) |
| `indvWrong` (IW) | Z | 1952 | 1949 (99.85) | 1078 | 1078 (100) |

Table 2 shows the total sample counts used for each figure panel. Note that as more conditions are added, the number of matching samples naturally decreases (and the variance typically increases).

*Table 2.* Sample counts and experimental details per figure panel.

| Fig. | Panel | Model | Count | Note |
|------|-------|-------|-------|------|
| 3 | b | DeepSeek-chat | 18,358 | Changed |
|   | c | GPT-5-mini | 9,642 | |
|   | d | DeepSeek-chat | 13,023 | Correctly revised |
|   | e | GPT-5-mini | 6,700 | |
| 4 | a | DeepSeek-chat | 7,500 | All wrongs |
|   | b | GPT-5-mini | 3,593 | |
|   | c | DeepSeek-chat | 4,176 | New wrongs |
|   | d | GPT-5-mini | 1,669 | |
| 5 | a | DeepSeek-chat | 3,459 | DC-A, UC-A |
|   | b | GPT-5-mini | 1,858 | |
|   | c | DeepSeek-chat | 214 | DC-A, UC-A, New wrongs |
|   | d | GPT-5-mini | 295 | |

### A.4.4. PROMPT VARIANTS FOR SOCIAL REVISION (SCENARIO TYPE SUFFIX)

Stage II uses a 3-message chat history with no system prompt: (1) user: original task prompt; (2) assistant: target previous answer; (3) user: reprompt (with or without peers). In addition, we evaluate two re-prompting variants, indicated by the suffix in scenario type: $*1$ (plain) and $*2$ (explicit reconsideration). For individual conditions, the scenario identifier is Z1 or Z2. Templates are shown in Figure 7.

---

**Stage II: Social Revision Prompts (no system prompt)**

**Message 1 (User)**
`<same baseline task prompt as Stage I>`

**Message 2 (Assistant)**
`{"grid":"<target_previous_grid>"}`

**Message 3 (User): Social variants**

```
(A1/B1/C1/D1)
Previously, you answered the puzzle as shown in your last message.
Now, here are the answers from 3 other solvers:
Answer from solver 1:  <peer_grid_1>

Answer from solver 2:  <peer_grid_2>
Answer from solver 3:  <peer_grid_3>
Output the answer you currently judge most likely to be correct.

Please output only your final answer in the same format as before.
```

---

```
(A2/B2/C2/D2)
Previously, you answered the puzzle as shown in your last message.
Now, here are the answers from 3 other solvers:
<same listing as above>
Carefully reconsider the original puzzle and all of these answers.
Please output only your final answer in the same format as before.
```

**Message 3 (User): Individual variants**

```
(Z1)
Previously, you answered the puzzle as shown in your last message.
Output the answer you currently judge most likely to be correct.
Please output only your final answer in the same format as before.
```

```
(Z2)
Previously, you answered the puzzle as shown in your last message.
Carefully reconsider the original puzzle.
Please output only your final answer in the same format as before.
```

*Figure 7.* Stage II social revision prompt templates (no system prompt). The A/B/C/D letters indicate the agreement count relative to the target (see Section A.4.2); the suffixes 1/2 indicate the reprompt variant.

## A.5. Parsing and Filtering

We parse outputs into integer grids. If parsing fails (e.g., due to invalid characters or a malformed grid), the sample/trial is marked invalid and removed from analysis. This filtering is applied to both the baseline pool from Stage I and the Stage II social-trial outputs.

**Label-aligned regularization.** When a parsed grid shape differs from the label shape, we apply the deterministic label-padding procedure: Crop extra rows/columns (top-left aligned), then pad missing rows/columns with value $-1$ until the label shape is matched. All distances are computed after this regularization.

## A.6. Outcome Measures and Distance Metrics

All distances (including the pixel-wise Hamming distance that are shown in the main manuscript, along with the two other auxiliary distance metrics) are computed after a deterministic label-aligned regularization step. (Invalid parses are filtered out and do not enter analysis.) Let $Y \in \mathbb{Z}^{H \times W}$ be the label grid and $\hat{Y}$ be a parsed prediction grid. If $\hat{Y}$ has a different shape, we apply label-padding: (i) crop extra rows/columns (top-left aligned) to match the label bounds, and then (ii) pad missing rows/columns with the constant value $-1$ until the shape equals $(H, W)$.

Each social trial yields a revised grid $G^{(1)}$ given target initial grid $G^{(0)}$ and the label grid $Y$. After the grid regularization, we measure:

- **Change indicator:** $\mathbb{I}[G^{(1)} \neq G^{(0)}]$.

- **Post-revision correctness:** $\mathbb{I}[G^{(1)} = Y]$ (exact match after regularization).

- **Peer-alignment type:** whether $G^{(1)}$ matches (i) the target-shown answer, (ii) one of the other shown wrong answers, or (iii) neither, used for stacked outcome breakdowns in plots.

and summarize them into a stacked bar for a clean visualization.

### A.7. Uncertainty Quantification (Bootstrap CIs)

For aggregated proportions (e.g., $\Pr[\text{CHANGED} = 1]$, class proportions in stacked outcome plots), we report nonparametric bootstrap percentile confidence intervals (Efron, 1979). Given $n$ trials and a binary indicator vector $x \in \{0, 1\}^n$, we resample $n$ trials with replacement $B = 1000$ times, compute the mean on each resample, and take the $(\alpha/2, 1 - \alpha/2)$ percentiles (with $\alpha = 0.05$) as the 95% interval. The same bootstrap CI is used in

### A.8. Computational cost and budget

A full evaluation "set" (fixed prompt variant, all experiment types and scenarios combinations) costs approximately 60M tokens total, including roughly 15M output tokens, which corresponded to approximately \$25 for DeepSeek-chat and \$75 for GPT-5 mini at the time this study was conducted.

## B. Additional distance metrics

In the main manuscript, we computed pixel-wise Hamming distance with crop and padding for the analysis. We report two additional distance metrics: the color-aware EMD-like matching distance (EM, for short) and the objective-level edit distance (edit, for short).

### B.1. Color-aware EMD-like Matching Distance

We compute an approximate Earth Mover's Distance (EMD)-like metric by solving a per-color assignment problem between pixel coordinate sets. Let $\mathcal{C}$ be the set of colors appearing in either $Y$ or $\hat{Y}$, excluding the ignore value $-1$. Let $D = \sqrt{H^2 + W^2}$ be the grid diagonal used for normalization. For each color $c \in \mathcal{C}$, define coordinate sets

$$L_c = \{(r, c) : Y_{rc} = c\}, \qquad P_c = \{(r, c) : \hat{Y}_{rc} = c\}.$$

We perform the following procedures:

**(i) Empty-side penalty.** If $L_c = \emptyset$ or $P_c = \emptyset$ but not both, we add a unit penalty per "mass":

$$\text{cost}_c \mathrel{+}= |L_c| + |P_c|, \quad \text{mass}_c \mathrel{+}= |L_c| + |P_c|.$$

If both are empty, we skip color $c$.

**(ii) Subsampling cap.** If $|L_c| > 200$, sample 200 points uniformly without replacement from $L_c$; likewise for $P_c$.

**(iii) Padded Hungarian assignment.** Let $n = \max(|L_c|, |P_c|)$ after subsampling. Construct a cost matrix $C \in \mathbb{R}^{n \times n}$ initialized as $C_{ij} = D$ (dummy cost). For each real label point $i \leq |L_c|$ and each real prediction point $j \leq |P_c|$, set

$$C_{ij} = \|(L_c)_i - (P_c)_j\|_2 \,.$$

Solve the minimum-sum assignment via the Hungarian method:

$$\text{cost}_c = \min_{\pi} \sum_{i=1}^{n} C_{i,\pi(i)}, \qquad \text{mass}_c = n.$$

Here, any unmatched mass is effectively matched to dummy points at a cost of $D$.

**(iv) Aggregation and normalization.** Let $\text{cost} = \sum_c \text{cost}_c$ and $\text{mass} = \sum_c \text{mass}_c$. If $\text{mass} = 0$, return 0. Otherwise:

$$d_{\text{emd}}(Y, \hat{Y}) = \frac{\text{cost}/\text{mass}}{D}. \tag{1}$$

Because of the empty-side penalty, the value is typically in $[0, 1]$, but can slightly exceed 1 in degenerate cases. Note that this metric is stochastic when subsampling is triggered.

**B.2. Object-level Edit Distance via Component Matching**

We define an object-level edit distance by extracting connected components per color and computing a minimum-cost matching with insertion/deletion penalties.

**(i) Object extraction.** For each grid, we extract 4-connected components separately for each color value, excluding $-1$. A component is the set of pixel coordinates reachable via 4-neighbor moves (up/down/left/right) staying within the same color. For each component $o$, we compute: area $a(o)$ (pixel count), bounding box $(r_{\min}, r_{\max}, c_{\min}, c_{\max})$, centroid $(\bar{r}, \bar{c})$, and a boolean shape mask $M(o)$ defined on the bounding box (top-left aligned).

**(ii) Pairwise edit cost between objects.** Let $D = \sqrt{H^2 + W^2}$. For a label object $o$ and a predicted object $p$, we compute

$$
\begin{aligned}
\text{cost}(o \to p) = & \; c_{\text{recolor}} \cdot \mathbb{I}[\text{color}(o) \neq \text{color}(p)] \\
& + c_{\text{move}} \cdot \frac{\|(\bar{r}_o, \bar{c}_o) - (\bar{r}_p, \bar{c}_p)\|_2}{D} \\
& + c_{\text{area}} \cdot \frac{|a(o) - a(p)|}{\max(a(o), a(p), 1)} \\
& + c_{\text{shape}} \cdot (1 - \text{IoU}(M(o), M(p))),
\end{aligned}
$$

where IoU is the intersection-over-union of the two boolean masks after padding them to a common size (top-left aligned). In our experiments, we use the unbiased equal weights for all operations:

$$
c_{\text{recolor}} = c_{\text{move}} = c_{\text{area}} = c_{\text{shape}} = 0.25.
$$

**(iii) Insert/delete costs and Hungarian matching.** Let $L = \{o_1, \ldots, o_{n_L}\}$ be label objects and $P = \{p_1, \ldots, p_{n_P}\}$ be predicted objects. Let $M = \max(n_L, n_P, 1)$. We build a square matrix $C \in \mathbb{R}^{M \times M}$ with the following:

$$
C_{ij} = \begin{cases}
\text{cost}(o_i \to p_j), & i \leq n_L, \; j \leq n_P, \\
c_{\text{delete}}, & i \leq n_L, \; j > n_P, \\
c_{\text{insert}}, & i > n_L, \; j \leq n_P, \\
0, & i > n_L, \; j > n_P,
\end{cases}
$$

with defaults $c_{\text{delete}} = c_{\text{insert}} = 1.0$. We then solve the minimum-sum assignment (Hungarian method) and compute the total cost $\text{Total} = \sum_{i=1}^{M} C_{i,\pi(i)}$. The final distance is normalized by $M$:

$$
d_{\text{edit}}(Y, \hat{Y}) = \frac{\text{Total}}{M}. \tag{2}
$$

## C. Results with auxiliary distance metrics

Here, we present all figures related to auxiliary distance metrics and reproduce the main figures using those metrics.

### C.1. Distance correlations

If our two auxiliary metrics were essentially identical to the existing pixel-wise Hamming distance, there would be no reason to adopt them as supplementary indicators. Therefore, as a sanity check, we calculated the cross-correlation among all three distance metrics for every misclassified example obtained in Stage I (Fig. 8). The three metrics showed moderate correlation—not completely independent, but not linearly dependent—confirming that they are sound and complementary indicators.

### C.2. Reproduction of the main figures

In Fig. 9, 10, we reproduce main figures: 3, 4, and 5, with $d_{\text{emd}}$ and $d_{\text{edit}}$. Overall, our findings are consistent across distance metrics, with few notable exceptions (such as a decreasing trend in Fig. 9(h) and an increasing trend in Fig. 11(f)). Some of

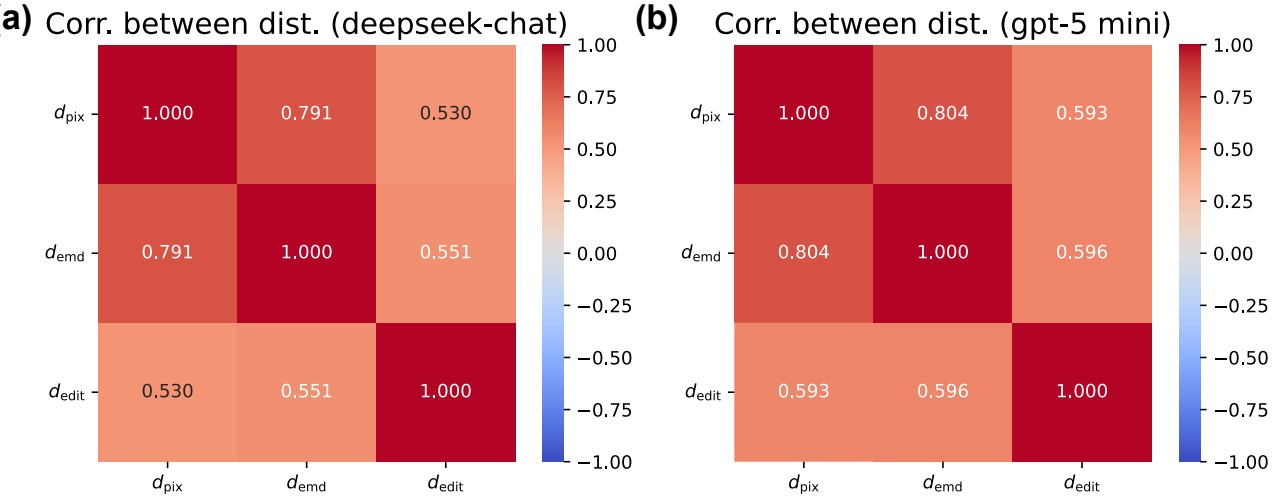

*Figure 8.* Correlations between three distance metrics, $d_{\text{pix}}$, $d_{\text{emd}}$, and $d_{\text{edit}}$, measured for all wrong answers from Stage I of (a) DeepSeek and (b) GPT-5 mini.

these may be statistical errors arising from aggregating only a very small number of samples into specific bin intervals (as a wide confidence interval would imply).

## D. Additional analyses

### D.1. Hamming distance-faithful subset of the results

Among the categories in ConceptArc, some are inherently unsuitable for hamming-distance-based answer assessment because they involve shifts/rotations or a non-standard answer format. To assess their effects, we removed several categories from ConceptArc that might not be a proper proxy for Hamming distances. Note that the result is critically dependent on the (rather subjective) choice of included and removed categories and re-drawn Figs. 3(b-e), 4, and 5. Note that this severely reduces the overall sample counts, and its effect is not uniform across all bins; hence, individual bins can end up with critically low sample counts, making them noisy and spiky.

- Included: AboveBelow, Center, CleanUp, CompleteShape, ExtendToBoundary, HorizontalVertical, InsideOutside, SameDifferent, Order, TopBottom2D, TopBottom3D

- Removed: Copy, Count, ExtractObject, FilledNotFilled, MovetoBoundary.

In Figure 12, we find that, despite suffering from low sample points, all of the figures and qualitative trends are mostly consistent with the full figure, except for panel j (corresponds to Fig.5b), DC-A (note that this plot has critically low sample points: the last bin only has 4 data points, previously 13). We believe that, in the future, our study can benefit from concurrent studies that are extending the ConceptARC dataset by adding more samples.

### D.2. Effect of soft imitation

To further address the concern about softer imitation, we conducted additional analyses (Fig. 13), as an extension of Fig. 4, to assess the effect of soft copying (partially imitating peers). Among newly generated wrong answers where the target's initial answer was farther from the ground truth than the best available wrong peer ($d_{\text{focal}} > d_{\text{peer, best}}$), making peer-surpassing non-trivial by construction. Even in this conservative subset, GPT-5 mini's newly generated wrong revisions surpass the best wrong peer in over $50\%$ of trials, whereas DeepSeek shows a relatively lower rate of outperformance. Since partial copying of peer structure cannot produce answers closer to the ground truth than the peer itself, this result provides direct evidence that contraction reflects genuine epistemic revision rather than softer imitation, especially for the higher-capability model.

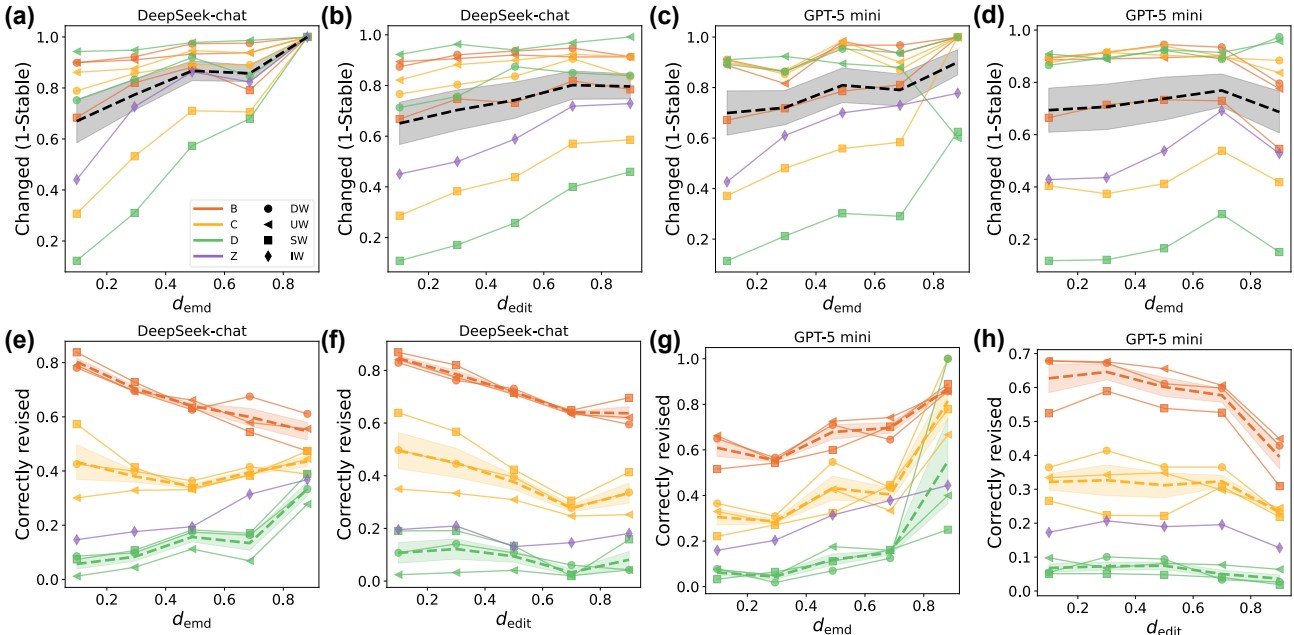

*Figure 9.* Distance-dependent revision behavior in wrong-target cases. (a-d) Probability of changed answers (equals to 1 minus the ratio of stable answers) among the wrong target cases for (a-b) DeepSeek-chat and (c-d) GPT-5 mini, by their target's $d_{emd}$ to the answer. (e-h) Probability of changed answers (equals to 1 minus the ratio of stable answers) among the wrong target cases for (e-f) DeepSeek-chat and (g-h) GPT-5 mini, by their target's $d_{edit}$ to the answer. Thin colored lines indicate individual experiment and scenario conditions, where the thick dotted line represents the average across (b-c) all conditions and (d-e) each scenario, with the shaded region indicating $\pm 1$ SEM.

DeepSeek's lower surpass rate (10–30%) suggests that the degree of genuine revision is modulated by model capability, consistent with the qualitative difference between the two models observed throughout the paper.

### D.3. Multi-round debate setting

One limitation of the current work is that the results primarily rely on single-step interactions, whereas multi-step social interactions are more common in real-world cases. Under conventional binary evaluation, the state space of an n-round debate grows moderately as $2^n$. Under our distance-aware framework, each round can produce a number of distinct answers in the continuous output space, so the number of unique trajectories grows much faster as $(d+1)^n$ ($d$: expected number of unique wrong answers). This is precisely why the paper focuses on single-step updates; however, we present our small-scale pilot results here to provide some limited multi-step verification.

For the multi-step verification, we conducted a 3-round pilot experiment to test whether the distance-aware dynamics persist across multiple rounds. We ran 3 rounds on GPT-5 mini under two conditions: diffWrong, with all three peers wrong (scenario D), the most pessimistic social signal and hence the harshest condition for contraction; and indvWrong (scenario Z, no peers) as a self-revision baseline. Peers are resampled independently from the Stage I pool at each round, holding social pressure constant across rounds. Note that this differs from a trial in which answers are sampled from the previous round, with new ones introduced; we leave this cumulative population effect for future work. We used all cases with problems that had more than 3 unique wrong answers (Scenario D: $n = 763$; Scenario Z: $n = 980$) across the full ConceptARC suite. We present our results in Fig. 14 and highlight three findings.

First, in the case of $d_{pix}$, Z shows a monotonic decrease while D shows a monotonic increase. This is not a contradiction; rather, it reflects our single-round finding that homogeneous wrong-peer pressure destabilizes correct answers (Section 4.1), thereby continuously injecting newly wrong cases into the population. Note that binary evaluation cannot distinguish these two forces. Secondly, the correct rate increases monotonically under Z but nearly stagnates under D, as all-wrong peer pressure counteracts accuracy gains each round. Finally, the most important point is that, even though the overall distance is

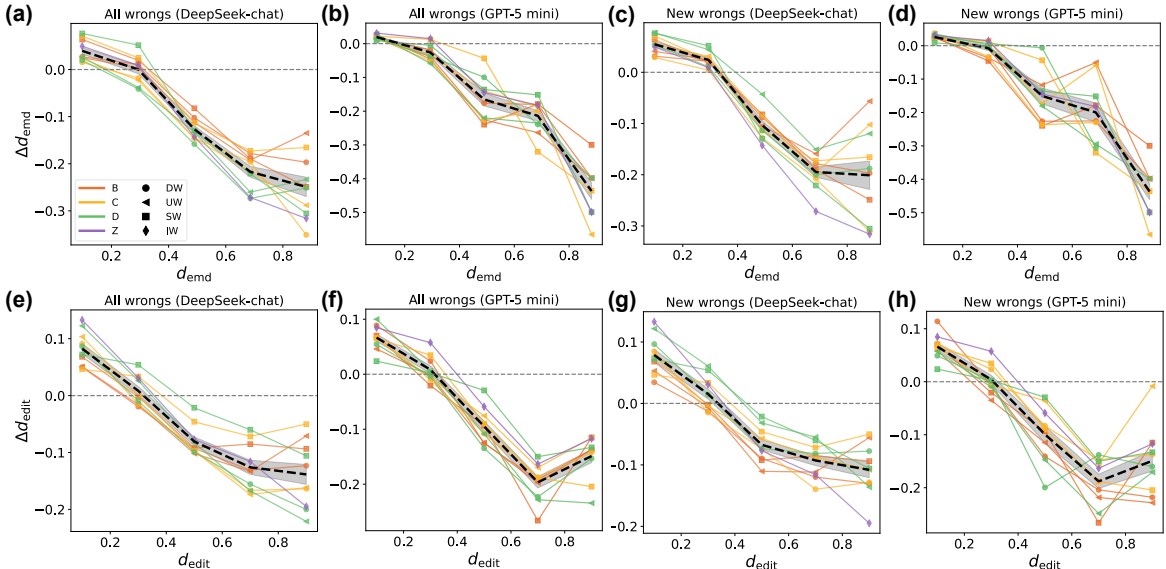

*Figure 10.* Distance contraction after revision for wrong target cases. (a-d) EM Distance change after revision ($\Delta d_{\text{emd}} = d_{\text{emd}}^{\text{after}} - d_{\text{emd}}^{\text{before}}$) from the wrong target cases, aggregated for (a-b) all wrong answers that are different from the target's initial answer, and (c-d) newly generated wrong answers different from the provided peer answers. (e-h) Edit distance change after revision ($\Delta d_{\text{edit}} = d_{\text{edit}}^{\text{after}} - d_{\text{edit}}^{\text{before}}$) from the wrong target cases, aggregated for (e-f) all wrong answers that are different from the target's initial answer, and (g-h) newly generated wrong answers different from the provided peer answers. Results are from (a, c, e, g) DeepSeek-chat and (b, d, f, h) GPT-5 mini. Thin colored lines indicate individual experiment and scenario conditions, where the thick black line represents the average, with the shaded region indicating $\pm 1$ SEM.

diverging and accuracy does not change, Per-revision contraction and its distance dependence persist across all three rounds, as in Fig. 4(c, d). The negative correlation between entering distance and replicates across all rounds, with D showing a stronger contraction magnitude than Z, might highlight a social effect (if it's quite far from the truth, then observing others, even though they're also wrong, can still help).

These results confirm that the distance-aware dynamics are not artifacts of one-shot exposure. They concretely illustrate the claim in the Discussion that iterative interaction can yield qualitatively different outcomes depending on whether contraction or correct-answer destabilization dominates: a distinction that is invisible under binary correctness. Full multi-round characterization, including modeling of convergence and oscillation, remains an imminent future direction.

### D.4. Additional model check: Gemini-2.5-flash-lite

We conducted an additional experiment to further investigate the qualitative difference between Fig. 3(d) and 3(e), namely, a different trend in the correction ratio as a function of the model, using another LLM. In the main manuscript, we explained and hypothesized that this difference stems from a systematic trend in baseline task performance, but the evidence wasn't conclusive, given only two models and no further analysis.

To further verify this, we reproduced Fig. 3(d, e) using Gemini-2.5-flash-lite (pass@1: 21.07%), a weaker model than both Deepseek (37.16%) and GPT (56.82%). We present the result in Fig. 15. Consistent with our hypothesis, this model also shows a decreasing correction rate as the correction rate increases, similar to Deepseek and contrary to GPT. The overall result (1) reconfirms our main findings with another model (which enhances the generality of the study), and (2) strengthens our hypothesis about the correction ratio.

### D.5. Prompt sensitivity

To assess the potential effect of different prompts during social interaction, we additionally test a variant of prompts by explicitly asking for "Carefully reconsider" rather than plainly asking for the best answer (see Fig. 7 for the exact prompts).

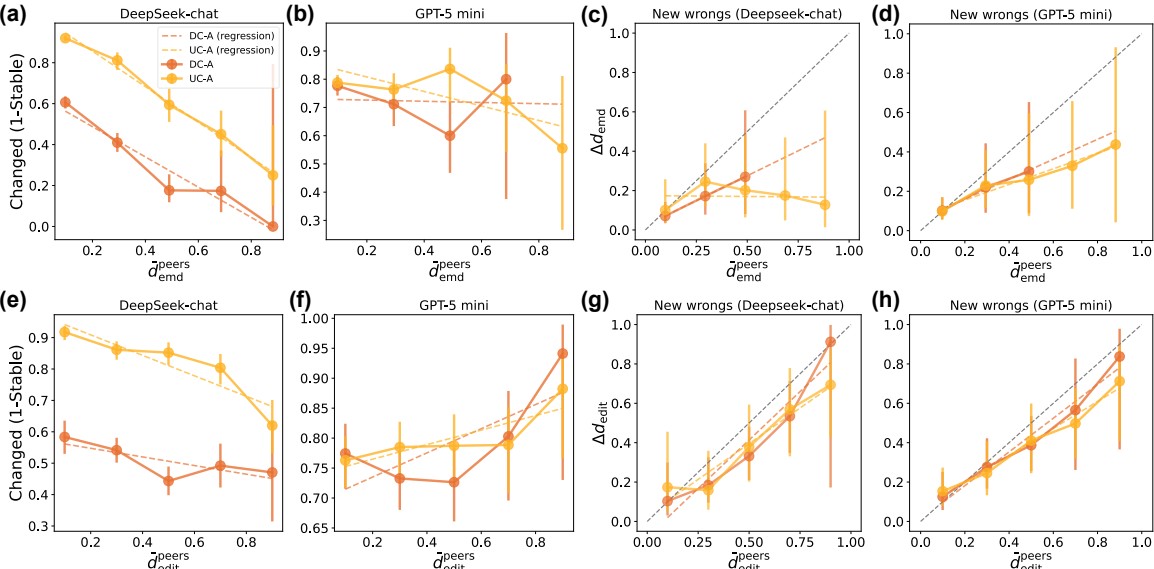

*Figure 11.* Distance-aware social influence for correct target cases under wrong peers (scenario A). (a-b) Probability of changing answer as a function of the mean peer EM distance $\bar{d}_{\text{emd}}^{\text{peers}}$ for all incorrect answers that differ from the target's initial answer. (c-d) Distance of newly generated wrong answers after revision. (e-f) Probability of changing answer as a function of the mean peer edit distance $\bar{d}_{\text{edit}}^{\text{peers}}$ for all incorrect answers that differ from the target's initial answer. (g-h) Distance of newly generated wrong answers after revision. Results are from (a, c, e, g) DeepSeek-chat and (b, d, f, h) GPT-5 mini. Error bars indicate 95% Wilson score interval.

As Fig. 16 shows, when asked to carefully reconsider, both models tend to overturn their belief and change the answer regardless of the initial answer of the target (except for a few cases where the original probability of change is too small, hence the ratio is not statistically meaningful). We find that this is a relatively uniform and systematic effect across all cases, and that further investigation into how differently engineered prompts affect distance-aware social dynamics would be a promising direction for future research.

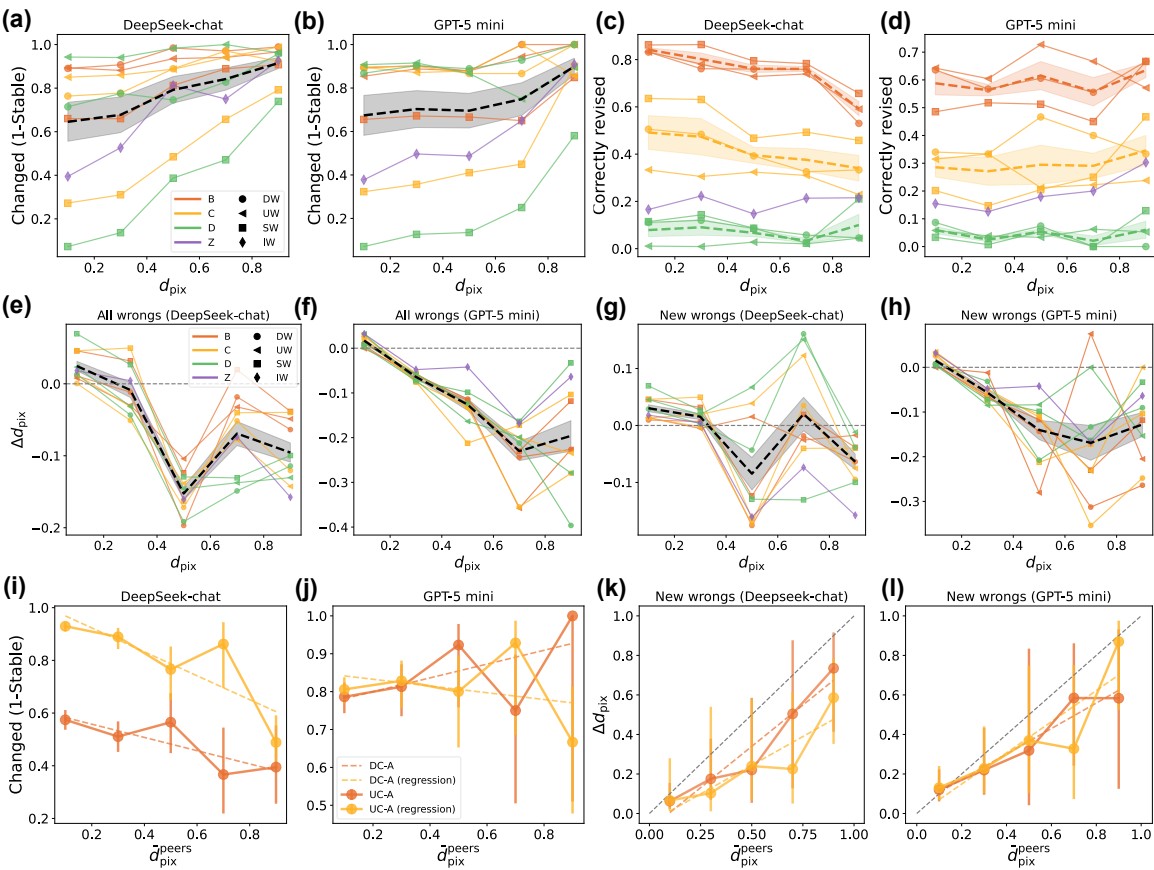

*Figure 12.* Reproduction of (a-d) Fig. 3(b-e), (e-h) Fig.4, and (i-l) Fig.5, with selected categories in ConceptARC where Hamming distance is a more reliable proxy for the semantic distance.

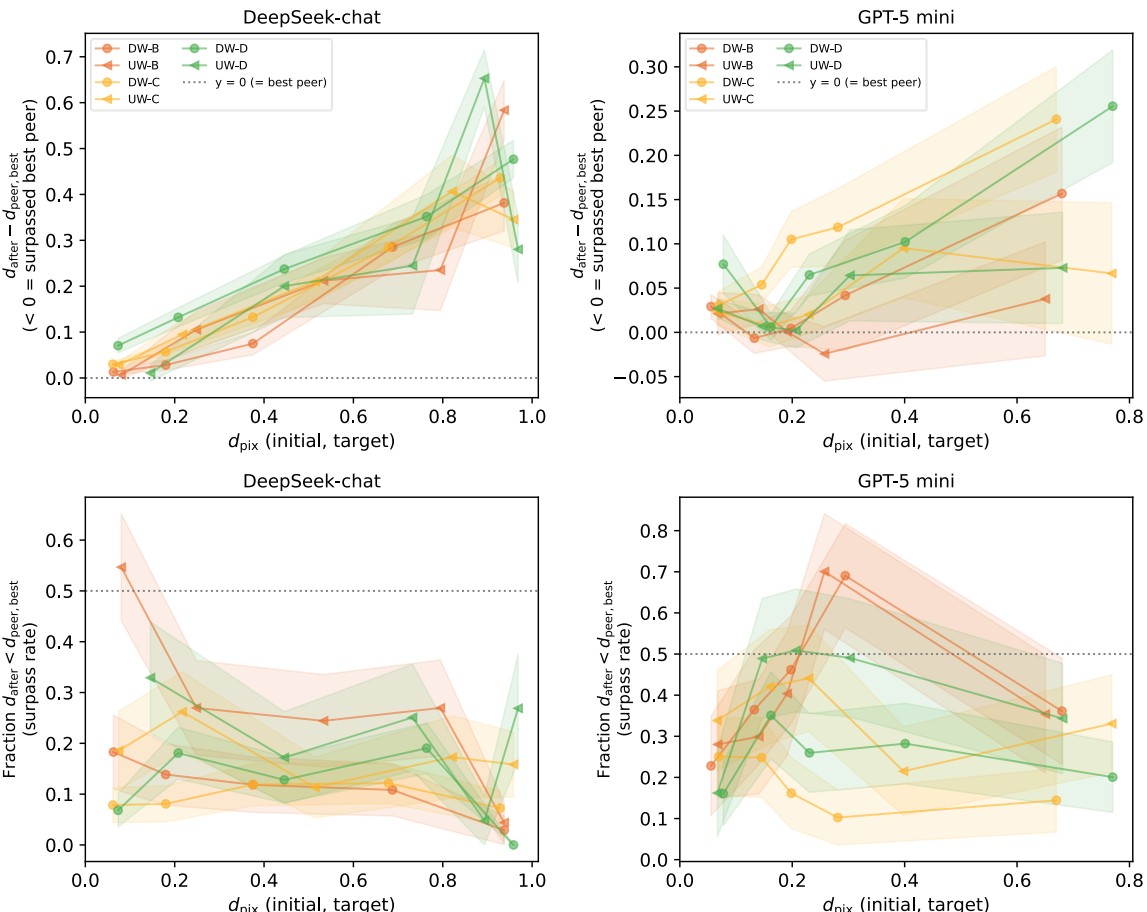

*Figure 13.* Peer copying analysis for newly generated wrong revisions in cases where the target's initial answer is farther from the ground truth than the best available wrong peer ($d_{\text{focal}} > d_{\text{peer, best}}$), ensuring that surpassing is non-trivial by construction. (a-b) Mean gap between the revised answer and the best wrong peer as a function of initial target distance. Values below zero indicate that the revised answer is closer to the ground truth than any of the provided wrong peers. (c-d) Fraction of trials in which the revised answer surpasses the best wrong peer. Results are shown for DeepSeek-chat (a, c) and GPT-5 mini (b, d). For conditions with mixed correct and wrong peers (scenario B and C), $d_{\text{best}}$ is computed over wrong peers only. Shaded regions indicate $\pm 1$ SEM.

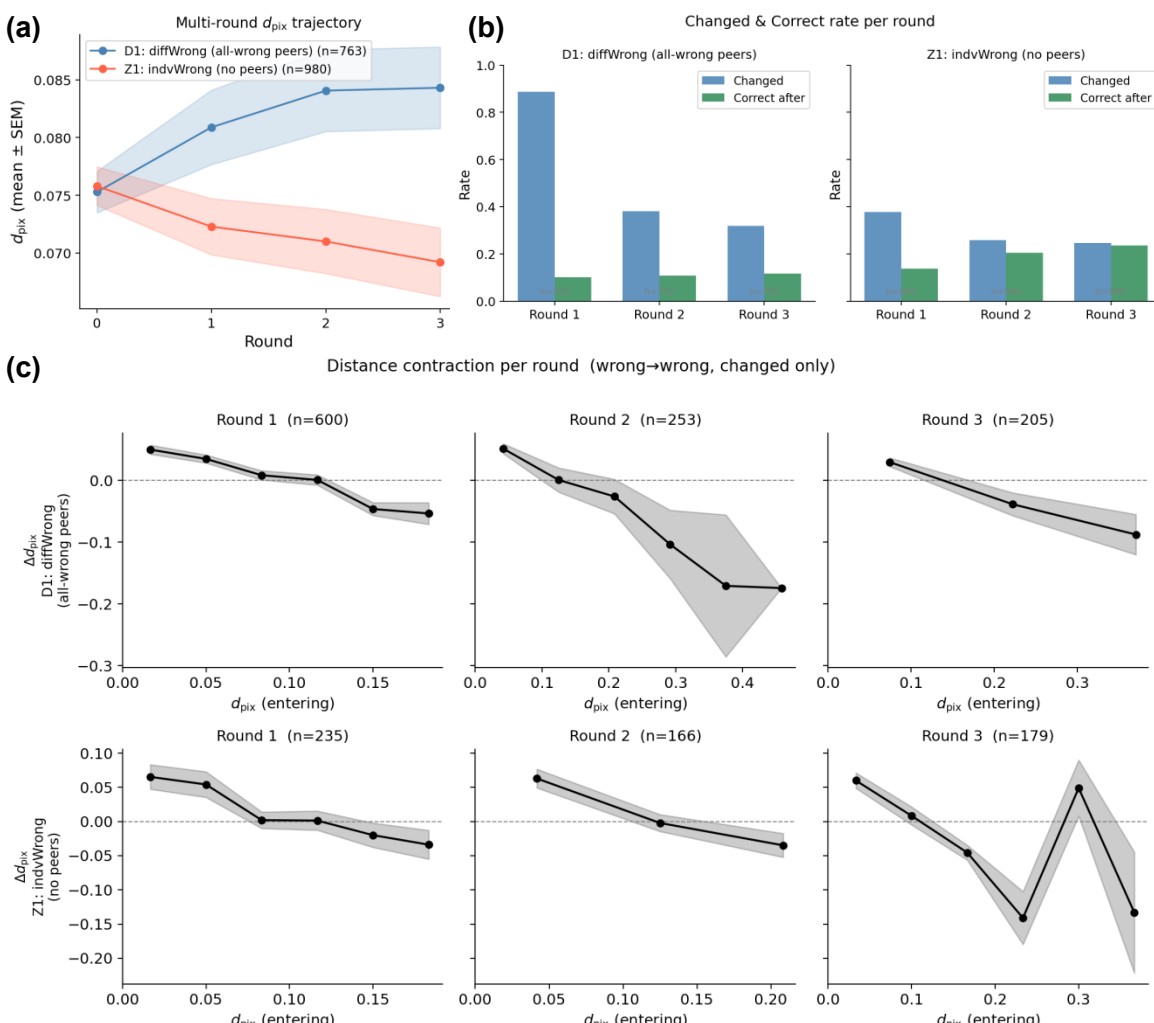

*Figure 14.* Result of multi-round experiment (GPT-5 mini, 3 rounds). (a) Mean $d_{pix}$ trajectory across rounds under scenario D (diffWrong, all-wrong peers) and Z (indvWrong, no peers). (b) Changed rate and correct rate per round for D (left) and Z (right). (c) Distance change after revision ($\Delta d_{pix} = d_{pix, after} - d_{pix, entering}$) as a function of entering distance, for wrong-to-wrong revised answers (changed only), shown separately for D (top) and D (bottom) across Rounds 1–3. Peers in D1 are resampled independently from the Stage I pool in each round, using the same scenario configuration. Shaded region: ±1 SEM.

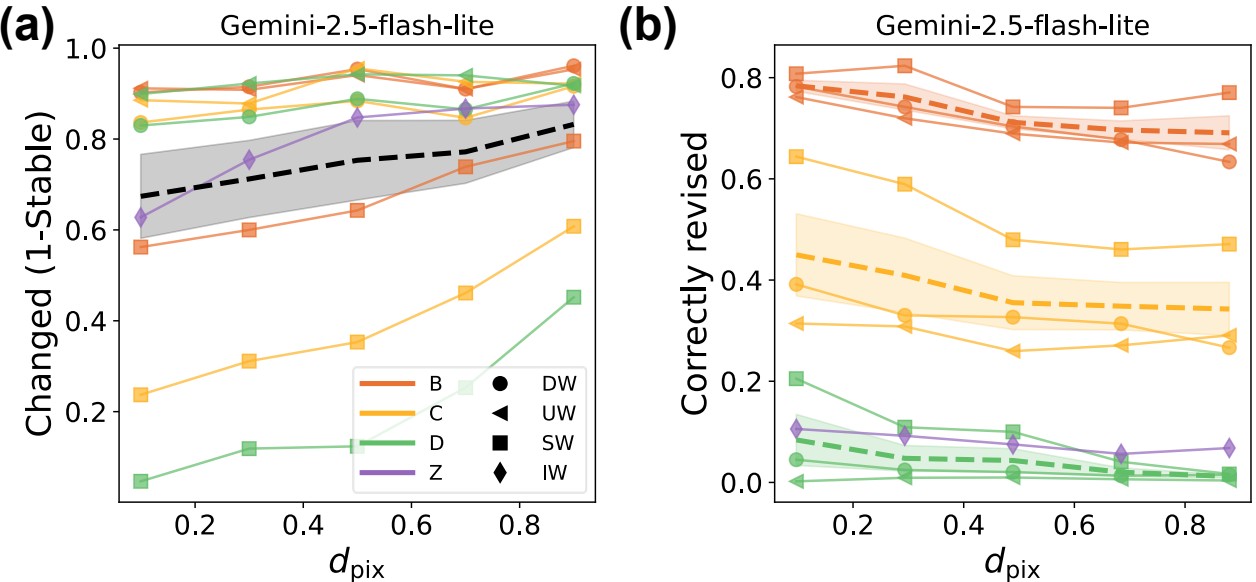

*Figure 15.* Distance-dependent revision behavior for wrong target cases, for Gemini-2.5-flash-lite (pass@1 accuracy 21.04%). (a) Probability of changed answers (equals to 1 minus the ratio of stable answers) among the wrong target cases by their target's pixel distances to the answer. (b) Probability of correctly revising answers from the wrong target cases. Thin colored lines indicate individual experiment and scenario conditions, where the thick dotted line represents the average across (a) all conditions and (b) each scenario, with the shaded region indicating ±1 SEM.

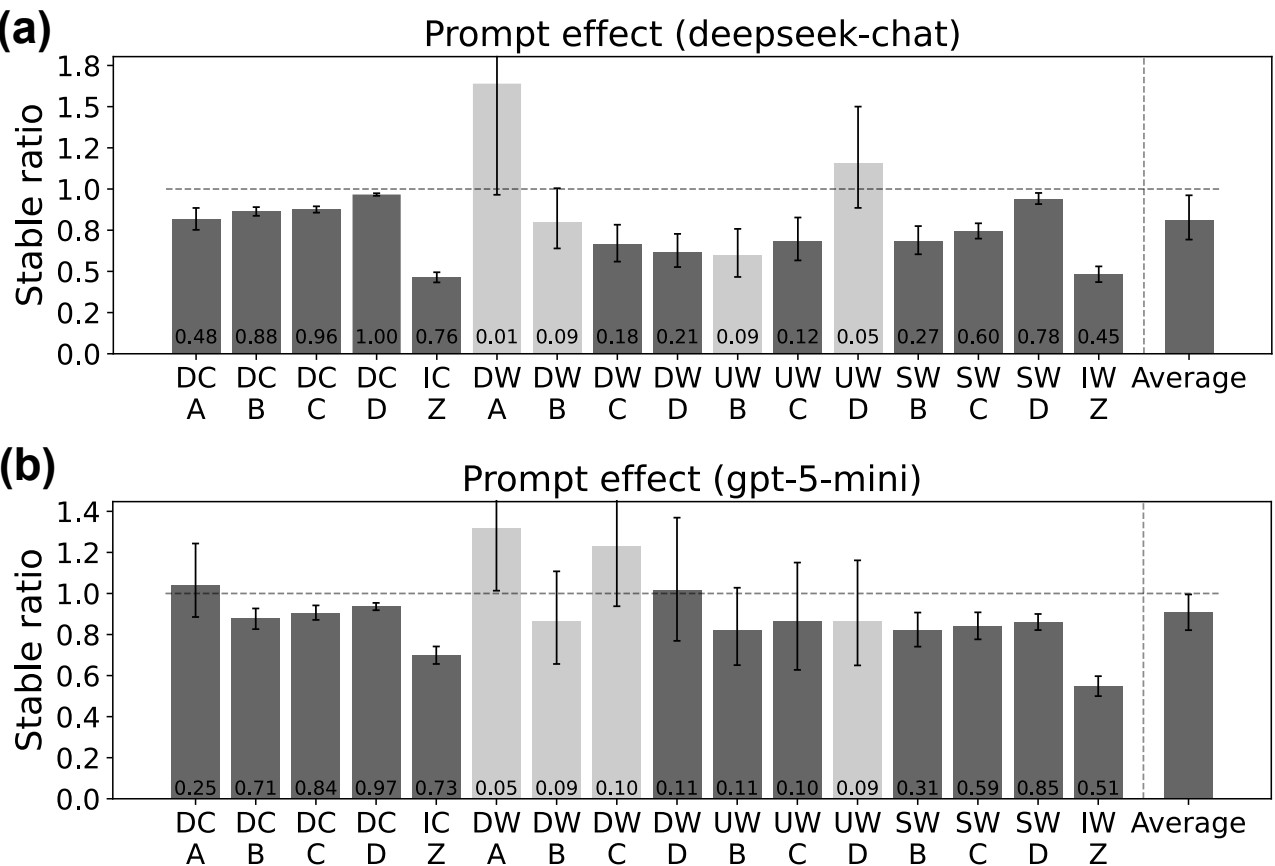

*Figure 16.* Ratio of stable revised answer between plain prompt (X1) and reconsideration prompt (X2) for (a) DeepSeek and (b) GPT-5 mini. The proportion of the stable for the plain prompt ($s_1$) is shown at the bottom of each bar, and scenarios where $s_1 < 0.1$ are colored light gray. Error bars indicate 95% CI by bootstrap methods with $1,000$ samples.

