# OpenReview forum: "Beyond Correctness: Distance-Based Social Dynamics of Multi-Agent Debate"
_ICML.cc/2026/Conference — ICML 2026 regular_

### Official Review · Reviewer_ruJZ · 2026-03-04

**Soundness:** 3
**Presentation:** 3
**Significance:** 2
**Originality:** 3
**Overall Recommendation:** 3
**Confidence:** 4

**Summary:**

Experimental work which evaluates how one-stage multi-agent debates between LLM agents can shift right or wrong decision of an agent to right or wrong direction under the social consensus pressure (Ash conformity), with quantitative evaluation of the opinion shifts and distances between the answers and ground truth, relying on the ConceptARC evaluation framework.

**Compliance With Llm Reviewing Policy:**

Affirmed.

**Final Justification:**

My concerns are partially resolved, but the remaining concerns are not easily addressed in a short rebuttal.

Reasons:
My points and questions are addressed, however the impact still appears too low given the limitations, and I see no novelty outlined.

In particular, my request for "Maybe considering some practical cases would increase the value of the work." is not addressed, I would recommend the authors either to consider practical applications and provide and demonstrate a working examples of applicability of the approach or defend some non-trivial novelty with a real practical impact.

Regarding missing the reproducible code in the original submission, I assume all authors were in equal conditions submitting their work by the same deadline, using either anonymized repositories (as it is done now past the submission deadline) or attaching their anonymised code right to submissions as supplementary material. That makes it a bit unfair to rate same way the papers which provide complete submissions on time and those which don't. This is not directly relevant to paper content, but just as recommendation to authors for future work.

Also, the technical note is that Q4 is not an answer to my question but copy&paste from reply to another reviewer.

So I am staying with my original assessment.

**Key Questions For Authors:**

- What is the reason of not providing reproducible code as it is required for scientific submission?

- How the images/grids in A3, Figure 6 were inserted in request if DeepSeek does not process images except for text extraction?

- If the images/grids were encoded as "concatenated texts" (literal strings) in JSON, what was the prompt explaining how to parse and generate JSON?

**Limitations:**

No explictit discission of limitations.

Limitations are implicitly mentined in Discssion section as directions for future work.

No obvious impact of this work, with some clarifications and options stated above.

**Strengths And Weaknesses:**

Strengths And Weaknesses

Strengths:
- Well designed and well performed experimental work
- Well structured paper
- Clear and elaborative discussion and analysis
- Findings are aligned with scope of knowledge in human psychology and sociology
- Use/study of modern LLMs (under assumption that use/study of LLMs is a must for scientific submission to be qualified)

Weaknesses
- Impact and value is not clear
- Finding and not clearly novel and applicable
- Paper is not comprehensible without of referring to appendix
- Reproducibility artifacts (code and data) are not included in submission
- Reproducibility instructions in appendix are not sufficient


Soundness:

>Is the submission technically sound?

Not that much. They mostly demonstrate the ability of LLMs to merge the items in the context.

>Are claims well supported (e.g., by theoretical analysis or experimental results)?

Yes. Findings are aligned with scope of knowledge in human psychology and sociology.

>Are the methods used appropriate?

Yes. Unless the objective for their use is not clear.

>If the paper includes theoretical results, are the proofs correct and based on reasonable assumptions?

Definitely.

>If the paper includes empirical results, are the experiments well-designed?

Paper is completely experimental, experimental design is perfect.

>Are the authors careful and honest about evaluating both the strengths and weaknesses of their work?

Somewhat. There is no Limitations section in the paper but authors have detailed discussion of the experimental results in the paper and further work in Discussion section. However, the value of the work and its applicability remains unclear.

Findings are aligned with scope of knowledge in human psychology and sociology, so the only novelty to me is that ability of a LLM architecture to merge the items in the context windows can emerge patterns known in human communications and reasoning. I guess that should be stated in the paper explicitly bit I am not sure about any other applications. Maybe considering some practical cases would increase the value of the work.

Presentation:

>Is the submission clearly written and well structured? (If not, please make constructive suggestions for improving its clarity.)

Generally, yes. Except the lack of reproducibility and need to lookup into Appendices while reading the paper.

>Is the overall narrative easy to follow?

Yes, except the above.

>Does the work properly position itself in the context of prior/concurrent literature and clearly discuss how it differs?

Yes. The main novelty is the Distance-Based assessment of difference in answers.

Significance:

>Does the paper address an important or relevant problem?

I am not sure about it and authors are fair about that in the Impact Statement.

>Does it advance understanding, capabilities, or practice in machine learning?

Yes, it confirms that LLM context operates accordingly to statistical evidence, as all machine learning does and all humans do.

>Could it influence future research or applications?

Potentially - yes, authors provide fair and reasonable directions for future work. It just has to be made clear what is the purpose of this work and its real-world or scientific value.

>Is the scope of impact broad or specialized, and is that appropriate for the contribution? Even if the improvements are modest or domain-specific, could they unlock new directions or provide practical utility?

It is in fact broad. I confirms that any LLM may be manipulated for good or for bad just exposing targeted evidence (for good or for bad, right or wrong), which is definitely a broad case for LLM security applications. I would suggest stating this in Impact Factor if the paper is accepted or if it is re-submitted in another venue.

Originality:

>Does the work provide new insights, deepen understanding, or highlight important properties of existing methods?

Limited. See above.

>Does the work introduce new tasks, methods, theory, data, or perspectives that advance the field in some dimensions?

Potentially, yes, but it has to be stated more clearly.

>Does this work offer a novel combination of existing techniques, and is the reasoning behind this combination well-articulated?

Yes, it offers the Distance-Based assessment of the decision shift and distance between the answer and the ground truth.

> Are the contributions clearly distinguished from closely related literature, and is the novelty well justified?

Yes, but the novelty is limited.

---

> ### Author Rebuttal · Authors · 2026-03-29
>
> Thank you for your thorough and detailed review of the manuscript, positive acknowledgement of the strengths of our study, and constructive comments. We carefully reviewed the raised issues and questions and provided the author's responses here. The changes will be faithfully reflected in the camera-ready version, as revisions are prohibited.
>
> ---
> # Q1. Impact on LLM research and Practical implications
> - First, we thank the reviewer for identifying the broader impact of our work and for the constructive suggestion.
> - We would like to clarify that our contribution goes beyond confirming that LLMs replicate known human social phenomena.
>   - (1) First, given the current active research and practical applications involving discussions among multiple LLM agents, studying **which aspects of human debate LLM discussions do and do not emulate** is significant in and of itself.
>   - (2) Our findings suggest that iterative pipelines may yield gains even when single-round accuracy improvements are negligible, because **even incorrect revisions systematically contract toward the ground truth**. This implies that majority voting, the most common method in MAD design, *should not be applied after just one round*, but rather after sufficient rounds of distance-reducing interaction for the maximum performance.
>   - (3) As the reviewer astutely notes, our results show that LLMs can be steered by selectively exposing them to near-plausible answers. Correct agents are most vulnerable to near-correct wrong peers (Fig. 5), not to clearly wrong ones. This asymmetry has **direct security implications and for adversarial robustness**: an attacker need not provide obviously wrong information; rather, a subtly **near-correct wrong answer** could be the most effective manipulation.
> - That said, we agree that the practical implications deserve more explicit treatment, and we will incorporate them into the Discussion and Impact Statement.
> ---
> # Q2. Reproducibility
> - First, it seems there was some misunderstanding regarding the provision of reproducible code, so I’d like to clarify this before moving on. As you are well aware, all manuscripts submitted to ICML must be anonymized for double-blind review, and even when a repository is provided, the URL cannot be included. Consequently, due to space constraints, the authors did not specify the data disclosure in the current submission and intended to make the full repository public in the camera-ready version. However, upon further investigation, **we realized there is a way to make the code publicly available via an anonymized URL.** It was our oversight not to explicitly state our intention to provide a codebase in the manuscript, as we were unaware of this method at the time; however, we wish to clarify that this was by **no means an attempt to withhold the code and data necessary for reproducibility**.
> - Therefore, to demonstrate our sincerity, we release the **entire codebase and an anonymized URL** in this rebuttal (https://anonymous.4open.science/r/socialLLM-EFEB/README.md). The current preview version of the repository includes all code used in this manuscript, and user-friendly manuals will be prepared before the camera-ready version is released.
> ---
> # Q3. Encoding and prompts
> - As described in the methods and provided prompts, all of the grids were encoded as concatenated strings, and the images in Figure 6 are for a **visualization purpose**. For clarification, we now inform this in the caption of Figure 6.
> - Also, we did not explicitly put any instructions about the format of JSON in prompts for simplicity, and both models were able to parse and generate JSON format with a few errors (we excluded them in further analysis). Since we did not fully disclose the sample counts for each condition, we added Tables 1 (https://anonymous.4open.science/r/socialLLM-EFEB/table1.png) and 2 (https://anonymous.4open.science/r/socialLLM-EFEB/table2.png) in the Appendix to disclose the exact number of valid samples for each experiment (see reply Q4 for reviewer BLZX (2nd) for more details).
> ---
> # Q4. Other important revisions
> - **"Where are the reproducible codes?"**: We are releasing our anonymized codebase (https://anonymous.4open.science/r/socialLLM-EFEB).
> - **"If it's only a single step, how can it be a debate?"**: We conducted an additional multi-step experiment (https://anonymous.4open.science/r/socialLLM-EFEB/FigR1.png) and added results to the manuscript. See the reply Q1 for reviewer agqd (3rd) for more details.
> - **"Some categories in ConceptARC are not semantically suitable for pixel-wise distances."**: We re-drew Figures 3-5 for selected categories where pixel distance is a more faithful semantic proxy (https://anonymous.4open.science/r/socialLLM-EFEB/FigR2.png) and confirmed that the results persist (see reply Q4 for reviewer spMB (1st) for more details).
> ---
> We hope our responses have helped you better understand our work, and we look forward to your positive feedback.
> Thank you!

---

> > ### Author Rebuttal · Reviewer_ruJZ · 2026-04-01
> >
> > My points and questions are addressed, however the impact still appears too low given the limitations, and I see no novelty outlined.
> >
> > In particular, my request for "Maybe considering some practical cases would increase the value of the work." is not addressed, I would recommend the authors either to consider practical applications and provide and demonstrate a working examples of applicability of the approach or defend some non-trivial novelty with a real practical impact.
> >
> > Regarding missing the reproducible code in the original submission, I assume all authors were in equal conditions submitting their work by the same deadline, using either anonymized repositories (as it is done now past the submission deadline) or attaching their anonymised code right to submissions as supplementary material. That makes it a bit unfair to rate same way the papers which provide complete submissions on time and those which don't. This is not directly relevant to paper content, but just as recommendation to authors for future work.
> >
> > Also, the technical note is that Q4 is not an answer to my question but copy&paste from reply to another reviewer.

---

> > > ### Author Response · Authors · 2026-04-05
> > >
> > > We thank the reviewer for their acknowledgment. As a final response, we address each point briefly.
> > >
> > > ---
> > > **On novelty.** We fully respect the reviewer's assessment on novelty, yet we would like to respectfully note that our main novelty claim rests on the distance-aware framework in MAD itself: reframing answer revision as continuous movement in the solution space rather than binary flips, as detailed in the Introduction and Discussion. This is distinct from prior MAD work, all of which treats revisions as categorical events, and has serious practical implications as follows.
> > >
> > > ---
> > > **On practical implications.** Our original rebuttal (Q1) explicitly addressed three practical cases: (1) iterative pipelines should apply majority voting after sufficient distance-reducing rounds rather than after one step; (2) near-correct wrong answers are the most effective adversarial signal for destabilizing correct agents; (3) the contraction finding implies that partially-correct solutions in open-ended tasks can be systematically improved through multi-agent interaction. We believe that all of this content can be immediately applied to MAD-related research and analysis of results that are currently taking place on a massive scale every day in academia and industry, including the domains of the legal, financial, and natural sciences.
> > >
> > > For a simple example, consider a multi-agent coding assistant where several LLM agents collaboratively construct a program. Under binary evaluation, if no agent produces a fully correct patch after a single round of debate, the round is deemed a failure, and one may deploy another batch of LLM agents to do it again (or even give up). Our framework reveals a different picture: even agents that remain incorrect after revision may have moved substantially closer to the correct patch in solution space, for instance, fixing the critical logic error but at the same time introducing a minor syntax mistake. This intermediate progress is invisible to binary evaluation but is precisely the signal that should trigger another round of revision rather than termination. Our contraction finding (Fig. 4) implies that such partially-improved wrong answers are the most likely to be corrected in the next round, since agents with smaller residual distance are more stable (Fig. 3b,c). A pipeline that monitors distance-to-solution rather than correctness alone could therefore decide when to stop iterating in a principled way: stopping too early discards recoverable progress, while stopping too late risks destabilization by social pressure (Fig. 5). This example illustrates that the practical value of our framework lies not in replacing existing MAD systems, but in providing richer signals for designing when and how to aggregate agent outputs.
> > >
> > > That said, we fully agree that an experiment using data from the practical domain would strengthen the paper, but it was (as the reviewer noted) beyond the scope of the short rebuttal. We hope that the framework presented in this study will inspire other researchers studying interactions among LLMs, leading to a variety of studies that also verify the framework's importance for real-world MAD applications using empirical data.
> > >
> > > ---
> > > **On Q4.** This section was intended to summarize significant revisions of the manuscript other than the reviewer's question, to help reviewers better understand the final status of the paper and overall quality. We apologize if it caused any confusion.
> > >
> > > ---
> > > **On code submission.** We acknowledge the reviewer's fair point and willingly take it as constructive feedback for future
> > > submissions.
> > >
> > > ---
> > > Again, we thank the reviewer for numerous constructive comments, and we believe that all reviewers significantly contributed to making this study more robust and comprehensive. We are looking for your positive final assessment.

---

### Official Review · Reviewer_agqd · 2026-03-13

**Soundness:** 3
**Presentation:** 3
**Significance:** 3
**Originality:** 3
**Overall Recommendation:** 4
**Confidence:** 3

**Summary:**

This paper states that standard evaluations of multi-agent debate are too coarse because they focus only on binary correctness or agreement. The authors show that answer revisions in LLM agents are distance-aware. That is, wrong answers that are farther from the ground truth are more likely to be revised, and many wrong-to-wrong revisions still move closer to the correct solution. At the same time, correct answers can be overturned when peer pressure comes from near-correct wrong answers. The result showhow social interaction shapes reasoning in multi-agent systems.

**Compliance With Llm Reviewing Policy:**

Affirmed.

**Key Questions For Authors:**

1. Why is one-step revision the right setting for the main claim, and do the authors have any evidence that the same pattern would persist in multi-round debate? The current paper studies only one-step revision, which is a limitation for a paper about multi-agent debate.

**Limitations:**

yes

**Strengths And Weaknesses:**

Soundness

- The paper's main claim is well supported. For example, the paper considers both wrong-target and correct-target cases, and also shows that many wrong-to-wrong revisions still move closer to the ground truth (Section 5).

- A majorweakness is that the paper only studies one-step revision. While it may be enough for the current claim, it is insufficient when it comes to the dynamics in a full multi-round debate.

- In addition, the experiments only use two models and one single benchmark. So it is still unclear how general the findings are.

Presentation

- The paper is clear overall.

- Section 3.2 is a bit hard to read because there are too many abbreviations at once (e.g., A/B/C/D, UC/DC, etc.). A summary table would help.

Significance

- The paper studies an important question. Many multi-agent debate papers focus only on final accuracy or agreement, so this work gives a more fine-grained view of what happens during revision.

- Another important result is that even incorrect revisions can still move toward the correct solution. I think this is interesting and useful.

Originality

- The analysis of local contraction toward the ground truth goes beyond the usual correct/incorrect view. This is the most original part of the paper.

- At the same time, the paper does not propose a new MAD method or a new model. So the originality is more in the evaluation angle and empirical insight, rather than in method design.

---

> ### Author Rebuttal · Authors · 2026-03-30
>
> Thank you for your thorough and detailed review of the manuscript, positive acknowledgement of the strengths of our study, and constructive comments. We carefully reviewed the raised issues and questions and provided the author's responses here. The changes will be faithfully reflected in the camera-ready version, as revisions are prohibited.
>
> ---
> # Q1. Multi-round evaluation
> - We thank the reviewer for this suggestion. Since this was requested by multiple reviewers, **we have conducted a 3-round pilot experiment to test whether the distance-aware dynamics persist across multiple rounds**.
> - Before we dive in, we would like to highlight the *inherent difficulty* of multi-round debate settings in our new framework. Under conventional binary evaluation, the state space of an n-round debate grows moderately as $2^n$. Under our distance-aware framework, each round can produce a number of distinct answers in the continuous output space, so the number of unique trajectories grows much faster as $(d+1)^n$ ($d$: expected number of unique wrong answers). This is precisely why the paper focuses on single-step updates; however, **we take the concern about multi-round validity seriously** and present our pilot results here.
> - We ran 3 rounds on GPT-5 mini under two conditions: diffWrong with all three peers wrong (scenario D), the most pessimistic social signal and hence **harshest condition for contraction**; and indvWrong (scenario Z, no peers) for a self-revision baseline. Peers are resampled independently from the Stage I pool at each round, holding social pressure constant across rounds. Note that this is different from a trial in which answers are sampled from the previous round, with new ones introduced: we leave this cumulative population effect for future work. We used all the cases with problems that have more than 3 unique wrong answers (D: n=763, Z: n=980) across the full ConceptARC suite.
> - We present our result (https://anonymous.4open.science/r/socialLLM-EFEB/FigR1.png), and three findings emerge.
>   - (a) Population-level $d_{\text{pix}}$ diverges: Z decreases monotonically (0.076→0.069), while D increases (0.076→0.084). This is not a contradiction; rather, it reflects our single-round finding that homogeneous wrong-peer pressure destabilizes correct answers (Section 4.1), thereby continuously injecting newly wrong cases into the population. Note that binary evaluation *cannot distinguish* these two forces.
>   - (b) Correct rate increases monotonically under Z (14%→21%→24%) but nearly stagnates under D (~11% flat), as all-wrong peer pressure counteracts accuracy gains each round.
>   - (c) The most important part is, **even though the overall distance is diverging and accuracy does not change, Per-revision contraction and its distance-dependency persists across all three rounds**, similar to Fig. 4(c, d). The negative correlation between entering distance and $d_{\text{pix}}$ replicates in all rounds, **with D showing stronger contraction magnitude than Z**, which might highlight the social effect (if it's quite far from the truth, then observing others, even though they're also wrong, can still help).
> - These results confirm that the distance-aware dynamics are not artifacts of one-shot exposure. They concretely illustrate the claim in the Discussion that iterative interaction can yield qualitatively different outcomes depending on whether contraction or correct-answer destabilization dominates: a distinction that is invisible under binary correctness. Full multi-round characterization, including modeling of convergence and oscillation, remains an imminent future direction.
> ---
> # Q2. Readability and Summary Table
> - Thanks for pointing out the readability in Section 3.2. We revised the manuscript to spread out cluttered abbreviations and generally clarify the wording.
> - Also, our **newly added Table 1** (See below) can help readers to summarize full experimental cases in one table.
> ---
> # Q3. Other important revisions
> - **"Where are the reproducible codes?"**: We are releasing our anonymized codebase (https://anonymous.4open.science/r/socialLLM-EFEB/README.md).
> - **"What are the exact sample counts for each experiment?"**: We added Table 1 (https://anonymous.4open.science/r/socialLLM-EFEB/table1.png) and 2 (https://anonymous.4open.science/r/socialLLM-EFEB/table2.png) to list all the valid sample counts after filtering the invalid responses. See the reply to question 4 for reviewer BLZX for more details.
> - **"Some categories in ConceptARC are not semantically suitable for pixel-wise distances."**: We re-drew Figures 3, 4, and 5 for selected categories where pixel distance is a more faithful semantic proxy (https://anonymous.4open.science/r/socialLLM-EFEB/FigR2.pdf) and indeed confirmed that the qualitative results persist (see reply Q4 for reviewer spMB (1st) for more details).
> ---
> We hope our responses have helped you better understand our work, and we look forward to your positive feedback. Thank you!

---

> > ### Author Rebuttal · Reviewer_agqd · 2026-04-04
> >
> > Thank you to the authors for the detailed rebuttal. I select (a) because my main concerns were adequately addressed. Overall, I think the rebuttal adequately addressed the concerns I raised. I am keeping my original score because it was already positive and, in my view, still matches the paper’s quality well.

---

> > > ### Author Response · Authors · 2026-04-05
> > >
> > > We thank the reviewer for their acknowledgment. As a final comment, we would like to present one more result that emerged while we were responding to another reviewer’s comments.
> > >
> > > ---
> > >
> > > In short, we conducted an additional experiment to further investigate the qualitative difference between Fig. 4(d) and 4(e), namely, a different trend in the correction ratio as a function of $d_{\text{pix}}$ between models, using **another LLM model**. We hypothesized that this difference stems from a systematic trend in baseline task performance, but the evidence wasn't conclusive, given only two models and no further analysis.
> > >
> > > To test this, we reproduced Fig. 4 (https://anonymous.4open.science/r/socialLLM-EFEB/FigR4.png) using Gemini-2.5-flash-lite (pass@1: 21.07%), a weaker model than both Deepseek (37.16%) and GPT (56.82%). Consistent with our hypothesis, this model also shows a decreasing correction rate as $d_{\text{pix}}$ increases, similar to Deepseek and contrary to GPT. The overall result (1) reconfirms our main findings with another model (which enhances the generality of the study), and (2) strengthens our hypothesis about the correction ratio. For a more detailed explanation, please visit the Reply Rebuttal Comment for reviewer spMD (1st).
> > >
> > > ---
> > >
> > > Again, we thank the reviewer for all the constructive comments, and we believe that all reviewers significantly contributed to making this study more robust and comprehensive. We are looking for your positive final assessment.

---

### Official Review · Reviewer_BLZX · 2026-03-13

**Soundness:** 2
**Presentation:** 2
**Significance:** 3
**Originality:** 3
**Overall Recommendation:** 4
**Confidence:** 3

**Summary:**

This paper studies multi-agent debate through a finer-grained lens than binary correctness, using ConceptARC as a benchmark where candidate outputs are 2D grids and can therefore be compared by distance to the ground truth. The authors construct controlled social scenarios in which a target LLM revises its answer after seeing peer answers, and analyze how revision probability and revision direction depend on whether the target is initially correct or wrong, the composition of peer answers, and the target or peer distance to the true solution. Across DeepSeek-chat and GPT-5 mini, the paper reports that wrong answers farther from the ground truth are more likely to be revised and often move closer to the truth even when still incorrect, while correct answers can be destabilized by socially presented near-correct wrong alternatives. The main message is that multi-agent interaction induces structured, distance-aware movements in solution space that are not visible under standard correctness-based evaluation.

**Compliance With Llm Reviewing Policy:**

Affirmed.

**Final Justification:**

The rebuttal addressed my main concern with a lot of experiments.

**Key Questions For Authors:**

Can you provide exact sample counts for each condition in Figure 2, Figure 3, Figure 4, and Figure 5?

Why should readers interpret the one-step answer-only setup as informative about “debate” rather than a narrower social revision primitive?

**Limitations:**

The task and definitions have limited scope.

**Strengths And Weaknesses:**

Soundness

W: The entire paper rests on one benchmark family, ConceptARC, and two proprietary or semi-proprietary LLMs. That is enough for an interesting case study, but not enough for the paper’s broader framing about multi-agent debate dynamics. The concern is that the central phenomenon may depend heavily on the geometry of grid outputs, the low-entropy answer space, and the specific prompting regime. A distance-aware story is much easier to tell when outputs are aligned grids with a natural Hamming metric than in open-ended reasoning tasks. The paper should either broaden the empirical scope or narrow the claims.

W: The social task here is one-step. It is more related to the concept of social learning instead of multi-agent debate. The title and framing emphasize “multi-agent debate,” but the actual experiment is closer to controlled social answer revision after seeing peer outputs.

W: The paper does not sufficiently disentangle peer copying from genuine reasoning, despite making a good attempt in Figure 4(c,d). Excluding exact peer matches is useful, but it still leaves open softer imitation effects, where the target partially copies structure from peers. Since the paper’s interpretation is epistemic improvement rather than imitation, a more detailed analysis of edit patterns or overlap with peer answers would strengthen the claim.


Presentation

S: The methods and experimental results are easy to follow.

W:  The paper includes many figures, but no compact table with model accuracies, valid sample counts, and number of social trials per condition.

Significance

S: The paper has a clear and focused empirical question, namely whether answer revision in multi-agent debate has structure beyond binary correct/incorrect outcomes. That is a worthwhile question, and the use of ConceptARC is well-motivated because the output space naturally supports quantitative distances. The research question is important and novel.

Originality

S: The idea of combining social interaction with pixel-level distance is novel.

W: The practical value of such social dynamics is limited. [1] provides some human-AI collaboration results to demonstrate the real-world value of the social learning study. Similar experiments should be added to make sure that such distance-based measurements have real-world meanings.

[1] Colas et al. Language and Experience: A Computational Model of Social Learning in Complex Tasks

---

> ### Author Rebuttal · Authors · 2026-03-30
>
> Thank you for your thorough and detailed review of the manuscript, positive acknowledgement of the strengths of our study, and constructive comments. We carefully reviewed the raised issues and questions and provided the author's responses here. The changes will be faithfully reflected in the camera-ready version, as revisions are prohibited.
>
> ---
> # Q1. Generality
> - We appreciate this concern and would like to clarify that the choice of ConceptARC is a deliberate methodological decision rather than a limitation to be overcome. The central contribution of this paper is to demonstrate that **a distance-aware perspective reveals structured dynamics that are invisible under binary evaluation**, and that this requires a benchmark in which a natural, interpretable distance metric exists between candidate solutions.
> - ConceptARC is, to our knowledge, one of the very few reasoning benchmarks that satisfy this requirement without additional training. The "low-entropy answer space" the reviewer mentions is not a confound: it is what makes fine-grained distance analysis tractable and interpretable.
> - That said, we take geometry-dependence seriously and note that our paper already includes **two additional distance metrics capturing distinct structural aspects beyond Hamming distance.** As shown in Fig. 8, these metrics are correlated but not redundant, and our main results replicate across all three (Figs. 9–11), reducing concerns that the findings are specific to Hamming geometry.
> - We agree that extending to other domains (such as coding and math) is a natural next step and will state this more explicitly, by slightly tone down the claim, present ConceptARC as a proof of concept for distance-aware analysis, and frame the broader implications as hypotheses for future work.
> ---
> # Q2. Multi-step experiment
> - To check whether our findings persist for multi-round debate, we conducted an additional experiment (https://anonymous.4open.science/r/socialLLM-EFEB/FigR1.png) and added results to the manuscript. Please see the reply Q1 to reviewer agqd (3rd) for more details.
> ---
> # Q3. Soft copying
> - The reviewer correctly notes that Fig.4 (c, d) leaves open **softer imitation**, partial structural overlap with peers. Indeed, soft copying definitely occurs, but there are signs that it is not the only mechanism driving the dynamics.
> - First, if contraction were mainly driven by soft imitation, it should be strongest when peers are closest to the ground truth. However, Fig. 4(c,d) shows contraction across all scenarios, including D (W3), where copying peers would not reliably move answers toward the solution. Its persistence even with all-wrong peers is therefore hard to explain by imitation alone.
> - Second, **we conducted an additional analysis that the reviewer requested**, using wrong cases that were *worse* than the best peer. If softer imitation were the main driver, we would expect a near-0 difference with peers and only rarely to outperform their best peers, which was not the case (https://anonymous.4open.science/r/socialLLM-EFEB/FigR3.png).
> ---
> # Q4. Valid sample counts
> - First, we thank the reviewer for pointing out this detail. Although some of the information the reviewer requested is listed in Section 4 and Appendix A.4.1, we find that the exact number of sample cases after 'excluding invalid outputs' is missing.
>    - Simply put, we have 1952 (DeepSeek) and 1078 (GPT) wrong cases and 480 correct cases, and we did a single social trial for all cases.
>    - But we removed invalid (JSON-unparsable) responses from further analysis. Further, we removed certain cases from scenarios in which Stage I produced too few wrong answers for the respective settings (for instance, scenario D for diffWrong requires at least 4 unique wrong answers). We find that the count of the remaining valid samples was not disclosed.
> - Therefore, **we summarized all of the valid sample count information** for Stage II in Table 1 (https://anonymous.4open.science/r/socialLLM-EFEB/table1.png) and 2 (https://anonymous.4open.science/r/socialLLM-EFEB/table2.png), and added descriptions in the manuscript. The mean error rates are 99.01% (DeepSeek) and 99.36% (GPT).
> ---
> # Q5. Practical value
> - We clarified our contribution for practical implications. Please visit reply to reviewer ruJz (4th), Q1.
> ---
> # Q6. Other important revisions
> - **"Where are the reproducible codes?"**: We are releasing our anonymized codebase (https://anonymous.4open.science/r/socialLLM-EFEB/README.md).
> - **"Some categories in ConceptARC are not semantically suitable for pixel-wise distances."**: We re-drew Figures 3, 4, and 5 for selected categories where pixel distance is a more faithful semantic proxy (https://anonymous.4open.science/r/socialLLM-EFEB/FigR2.png) and confirmed that the results persist (see reply Q4 for reviewer spMB (1st) for more details).
> ---
> We hope our responses have helped you better understand our work, and we look forward to your positive feedback. Thank you!

---

> > ### Author Rebuttal · Reviewer_BLZX · 2026-04-02
> >
> > Thanks for all the experiments. My main concerns are resolved and I raised my score accordingly.

---

> > > ### Author Response · Authors · 2026-04-05
> > >
> > > We thank the reviewer for their acknowledgment and for the positive adjustment to the score. As a final comment, we would like to present one more result that emerged while we were responding to another reviewer’s comments.
> > >
> > > ---
> > >
> > > In short, we conducted an additional experiment to further investigate the qualitative difference between Fig. 4(d) and 4(e), namely, a different trend in the correction ratio as a function of $d_{\text{pix}}$ between models, using **another LLM model**. We hypothesized that this difference stems from a systematic trend in baseline task performance, but the evidence wasn't conclusive, given only two models and no further analysis.
> > >
> > > To test this, we reproduced Fig. 4 (https://anonymous.4open.science/r/socialLLM-EFEB/FigR4.png) using Gemini-2.5-flash-lite (pass@1: 21.07%), a weaker model than both Deepseek (37.16%) and GPT (56.82%). Consistent with our hypothesis, this model also shows a decreasing correction rate as $d_{\text{pix}}$ increases, similar to Deepseek and contrary to GPT. The result (1) reconfirms our main findings with another model and (2) strengthens our hypothesis about the correction ratio. For a more detailed explanation, please visit the Reply Rebuttal Comment for reviewer spMD (1st).
> > >
> > > ---
> > >
> > > Again, we thank the reviewer for all the constructive comments, and we believe that all reviewers significantly contributed to making this study more robust and comprehensive. We are looking for your positive final assessment.

---

### Official Review · Reviewer_spMB · 2026-03-13

**Soundness:** 3
**Presentation:** 3
**Significance:** 3
**Originality:** 3
**Overall Recommendation:** 4
**Confidence:** 4

**Summary:**

This paper argues that evaluating multi-agent debate with binary correctness misses most of the interesting dynamics. Using ConceptARC, a grid-reasoning benchmark where you can measure continuous distances between answers and ground truth, they show that LLM agents revise in a distance-dependent way: agents farther from the solution are more likely to change, and even revisions that stay wrong systematically contract toward the correct answer. On the flip side, correct agents are most vulnerable to near-correct wrong peers, not wildly off ones. The contraction holds even for genuinely new answers (not copied from peers), and the stronger model shows cleaner behavior. The message is that there's structured, gradual epistemic movement happening beneath the surface that binary metrics simply cannot capture.

**Compliance With Llm Reviewing Policy:**

Affirmed.

**Key Questions For Authors:**

1. Have you run even a small-scale experiment with 2-3 rounds of revision?

2. The opposite trends in correction probability between DeepSeek-chat and GPT-5 mini (Fig. 3d vs 3e) are striking. Do you have any intuition for why this happens? Have you considered testing a third model to see which pattern is more typical?

3. For tasks involving spatial shifts or rotations, Hamming distance can be large even when the answer is semantically close. Did you check whether your main findings hold when restricted to task categories where pixel distance is a more faithful semantic proxy?

4. Have you looked at whether token-level log-probabilities or self-reported confidence correlate with the distance-dependent revision patterns?

**Limitations:**

Yes. The authors are upfront about the single-round restriction, the two-model scope, and the semantic limitations of Hamming distance. The societal impact statement is minimal but reasonable given the nature of the work.

**Strengths And Weaknesses:**

Strengths

1. The core idea is fresh. Most MAD papers stop at "does debate help accuracy?" Reframing revision as continuous movement in solution space is a simple but powerful shift. Using ConceptARC is clever since it's one of the few benchmarks where a natural distance metric exists without needing a learned verifier.

2. The experimental design is careful. The scenario matrix covers the right axes of variation, and the authors address the obvious confound (contraction from peer-copying) by separately analyzing newly generated answers. Error bars are reported throughout, and auxiliary metrics in the appendix add robustness.

3. The paper reads well. The progression from categorical (Section 4) to distance-based analysis (Section 5) builds naturally, and the connections to classic social psychology feel earned, not forced.

4. The contraction finding could change how people design and evaluate multi-agent systems. If wrong-but-improving revisions are common, iterative pipelines might work even when single-round accuracy gains look negligible.

Weaknesses

1. Two models is thin for general claims. Worse, the qualitative differences between them (e.g., opposite correction trends in Fig. 3d vs 3e) actually undermine generality. A third model from a different family would help considerably.

2.The discussion leans heavily on multi-round implications (iterative contraction, convergence), but these remain speculative without even a small pilot over 2-3 rounds on a subset of tasks.

3. Hamming distance is semantically shallow. For tasks involving spatial transformations, two answers can be close in meaning but far in pixel space. Promoting at least one auxiliary metric to the main text would strengthen confidence in the findings.

4. The paper documents what happens but not why. Is distance-dependent revision related to the model's own token-level confidence? Even correlational analysis would add depth.

5. The framework requires a natural output-space distance metric. The paper doesn't discuss how these findings might transfer to math, code, or text where such metrics are much harder to define.

---

> ### Author Rebuttal · Authors · 2026-03-30
>
> Thank you for your thorough and detailed review of the manuscript, positive acknowledgement of the strengths of our study, and constructive comments. We carefully reviewed the raised issues and questions and provided the author's responses here. The changes will be faithfully reflected in the camera-ready version, as revisions are prohibited.
>
> ---
> # Q1. Multi-step experiment
> - To check whether our findings persist for multi-round debate, we conducted an additional experiment (https://anonymous.4open.science/r/socialLLM-EFEB/FigR1.png) and added results to the manuscript. Please visit the reply Q1 for reviewer agqd (3rd) for more details.
> ---
> # Q2. What makes Fig. 3(d, e) different?
> - Indeed, the opposite trends between the models (Fig. 3d vs. 3e) are striking, and we offer the following interpretation, which we have added to the manuscript.
> - We argue that **this divergence is not a threat to generality, but rather an additional finding** that is consistent with a capability-modulated view of social revision. Our central claims, that revision likelihood increases with initial distance (Fig. 3b,c) and that wrong-to-wrong revisions systematically contract toward the ground truth (Fig. 4), hold robustly for both models. The divergence in Fig. 3d vs. 3e concerns a **secondary question: whether distance-dependent revision translates into correct answers**.
> - The key moderating variable is **baseline task performance**. GPT achieves 56.82% accuracy on ConceptARC, compared to 37.16% for DeepSeek, indicating a stronger underlying grasp of the rule-induction structure. For a capable model, a large initial distance signals a fundamental misunderstanding, and a second opportunity allows it to re-engage and recover the correct rule. DeepSeek, with weaker performance, moves toward peers without fully recovering the correct solution, yielding lower and distance-independent correction rates. **This is further supported by peer overlap analysis** (see reply Q3 for reviewer BLZX (2nd) for more details): GPT's revisions surpass the best wrong peer in over 50% of cases, while DeepSeek's surpass rate remains below 30%, confirming GPT's stronger independent reasoning even during social revision.
> - We agree that adding 3rd model would strengthen generality claims and will include it as a prioritized direction for future work. We note, however, that the two models already span a meaningful range of capabilities (37% vs. 57%), and the directional difference between them is systematic and interpretable rather than arbitrary.
> ---
> # Q3. Selected categories
> - As the reviewer suggested, we removed several categories from ConceptArc that might not be a proper proxy for Hamming distances that have shift/rotations or a non-standard answer format. Note that the result is critically dependent on the subjective choice of which categories to remove. Also, this severely reduces sample counts, and its effect is *not uniform* across all bins; hence, individual bins can have low sample counts and become noisy.
>   - Removed (5 out of 16): Copy, Count, ExtractObject, FilledNotFilled, MovetoBoundary
> - We find that, despite suffering from low sample counts, **all of the figures and qualitative trends are mostly consistent** with the original, except for panel j (corresponds to Fig.5b), DC-A (note that this plot has critically low sample points: the last bin only has 4 data points, previously 13). We believe that, in the future, our study can benefit from concurrent studies that are extending the ConceptARC dataset by adding more samples.
> ---
> # Q4. Misc.
>  -  We revised the manuscript to **promote auxiliary metrics more**, and where Hamming distance could fail.
> - We thought about asking for self-reported confidence, but token-level confidence is an interesting idea. We need to re-run the entire social trial to check it at this time (since we didn’t ask for confidence or save the token-level logits), but we thank the reviewers for these valuable suggestions for our future work and will add a section on confidence-based correlation to the Discussion.
> - We agree that broadening to other domains (such as coding and math) is a natural next step, and there are already several potential ways and attempts to define 'distances' in those domains. We will make this more explicit as a future direction in the Discussion.
> ---
> # Q5. Other important revisions
> - **"Where are the reproducible codes?"**: We are releasing our anonymized codebase (https://anonymous.4open.science/r/socialLLM-EFEB/README.md).
> - **"What are the exact sample counts for each experiment?"**: We added Table 1 (https://anonymous.4open.science/r/socialLLM-EFEB/table1.png) and 2 (https://anonymous.4open.science/r/socialLLM-EFEB/table2.png) to list all the valid sample counts after filtering the invalid responses. See the reply Q4 for reviewer BLZX (2nd) for more details.
> ---
> We hope our responses have helped you better understand our work, and we look forward to your positive feedback. Thank you!

---

> > ### Author Rebuttal · Reviewer_spMB · 2026-04-03
> >
> > After the rebuttal, we maintain our previous evaluation of "Weak acceptance". We believe that the core weaknesses of the work, namely the need for more experiments with a wider range of models and a deeper exploration of the underlying mechanisms, will require a greater amount of work beyond what can be addressed in a simple rebuttal.

---

> > > ### Author Response · Authors · 2026-04-05
> > >
> > > We thank the reviewer for their acknowledgment.
> > >
> > > ---
> > > In fact, the authors were also strongly intrigued by the question raised by the reviewer: *What is the main reason for the qualitative differences between Fig. 4(d) and 4(e)*? In the rebuttal, we hypothesized that it's a **systematic trend arising from inherent task performance**: a stronger model can identify obvious mistakes that cause high distances more easily and thus shows a higher correction rate for large distances (as in Fig. 4(e) for GPT), a weak model is incapable of do so and suffers from bad original guess (as in Fig. 4(d) for Deepseek).
> > >
> > > Indeed, to confirm whether the exact cause of this phenomenon is as proposed by this hypothesis, we would need to test a wider range of models with varying baseline performance. But this goes beyond the scope of a short rebuttal, as the reviewer noted. Nevertheless, since we also wanted to check the validity of the hypothesis, we were able to conduct a quick additional experiment that adds one more piece of evidence using another LLM model.
> > >
> > > ---
> > > Here, we reproduced Fig. 4 using Gemini-2.5-flash-lite (https://anonymous.4open.science/r/socialLLM-EFEB/FigR4.png), a lightweight proprietary model that is faster and more cost-effective, though less capable, with a pass@1 accuracy of 21.07% (compared to 37.16% (Deepseek) and 56.82%(GPT) for the ConceptArc task). If our hypothesis is true, **this (worst) model needs to also show a decreasing correction ratio as the $d_{\text{pix}}$ increases**, similar to (worse-performing) Deepseek. Due to time constraints, we conducted this experiment with $10$ trials of Stage I rather than $20$, but the number of unique wrong cases (2,928) was much higher because of the low accuracy.
> > >
> > > From panel (a), we confirmed that the results of this model also show a consistently high rate of answer changes as $d_{\text{pix}}$ increases, thereby revalidating one of the key findings of our study. More importantly, panel (b) shows that its correction rate indeed decreases as $d_{\text{pix}}$ increases, similar to Deepseek and contrary to GPT. This additional test case is more of a supporting evidence than proof of the hypothesis or a thorough analysis of the underlying mechanism, but it definitely strengthens our original hypothesis. We hope that the framework presented in this study will inspire other researchers studying interactions between LLMs, leading to a series of studies to determine whether these phenomena can indeed be replicated with other models and under different conditions (including a replication of Fig. 4 using models more powerful than GPT-5 mini).
> > >
> > > ---
> > > Again, we thank the reviewer for all the constructive comments, and we believe that all reviewers significantly contributed to making this study more robust and comprehensive. We are looking for your positive final assessment.
> > >
> > > p.s. We found that in the previous rebuttal, we forgot to include the result figure for **Q3. Selected categories**. We attach it here (https://anonymous.4open.science/r/socialLLM-EFEB/FigR2.png).

---

### Decision · Program_Chairs · 2026-04-30

**Decision:**

Accept (regular)

**Comment:**

This is an innovative paper regarding multi-agent debates. The reviewers pointed out a number of issues, but they were addressed in the rebuttal (though one reviewer remains unconvinced). Altogether, I believe that a paper that does something original---as is in this case---should be allowed some slack in quality of evaluation. One cannot do everything in one paper.